# Aperiodic 1/f noise drives ripple activity in humans

Frank J. van Schalkwijk [1] ✉ & Randolph F. Helfrich [2,3,4] ✉

Sharp-wave ripples (SWR) are central for cognition and hallmark sleep in the rodent hippocampus. Recently, ripple-like activity was also observed in the human hippocampus and neocortex during wakefulness and sleep. However, ripple detection across brain states and cortical regions remains challenging. We demonstrate that putative ripples largely index noise originating from region-, state-, and demand-dependent modulation of cortical background activity. We establish the noise sensitivity for five common detection algorithms across three intracranial EEG studies during sleep and cognitive engagement. On average, 77% of awake ripples in the medial temporal lobe, including the hippocampus, reflect false positives within the $1/f^x$ noise floor. We also report task-related $1/f^x$ modulations that lead to spurious ripple activity, and demonstrate scenarios where ripple detections are less impacted by noise. Our results offer a simulation-based approach to estimate the false positive rate and demonstrate the importance of $1/f^x$ activity for state- and context-dependent cortical processing.

Hippocampal sharp-wave ripples (SWR) constitute one of the central building blocks underlying cognition and memory in the mammalian brain[1]. SWR reflects synchronized volleys of population activity in the frequency range from ~80–200 Hz that preferentially occur during slow wave sleep (SWS). During SWR, population activity patterns are reactivated and repeated in a process known as replay[2,3], which is thought to facilitate neuroplasticity underlying long-term memory storage[4–6]. While SWR were mainly described in the CA1 region of the rodent hippocampus during sleep, recent studies described ripple-like signatures in the human hippocampus[7–12] as well as in neocortex[13–16], and have additionally been observed during wakefulness[10,16,17]. Especially, wake ripples have recently been implicated in several cognitive functions, including memory retrieval, navigation, and planning[10,18,19]. However, it has been fiercely debated whether awake ripples recorded from the neocortex constitute true functional homologs of the hippocampal SWR[20]. In humans, ripples were mainly reported in epilepsy patients undergoing presurgical intracranial electroencephalography (iEEG) monitoring with implanted depth or subdural grid electrodes. Identification of ripples in this patient population is hampered by several factors, including the size of clinical macro-electrodes, which

often span several hippocampal subfields, as well as pathological electrophysiological signatures of epilepsy, such as interictal spikes, sharp waves, and high-frequency oscillations[21–24]. Recently, ripple-like signatures and replay were also reported from non-invasive source imaging using magnetoencephalography (MEG) in non-epileptic volunteers[25,26]; however, it remains unclear if low-amplitude, high-frequency activity in the medial temporal lobe (MTL) can be precisely localized with MEG. Currently, there is no consensus about which features and characteristics define a ripple in humans and to what extent these signatures actually correspond to the canonical hippocampal SWR[20].

In a largely unrelated line of inquiry, it had recently been shown that electrophysiological 'background' activity, also termed aperiodic activity for the lack of a unique periodicity, indexes different brain states and cognitive demands. Aperiodic activity is also called 1/f noise as it follows a $f^x$ scaling law. We defined 1/f noise as $1/f^x$, which yielded a positive exponent. We adopted the common convention to report the exponent in slope form as the negative exponent[27,28]. Spectral exponents typically range from −1 to −2 during cognitive engagement[29,30], from −2 to −4 during sleep[31–35], and have values below −4 during

[1]Hertie-Institute for Clinical Brain Research, Center for Neurology, University Hospital Tübingen, Tübingen, Germany. [2]Department of Psychology, Yale University, New Haven, CT, USA. [3]Wu Tsai Institute, Yale University, New Haven, CT, USA. [4]Interdisciplinary Neuroscience Program, Yale University, New Haven, CT, USA. ✉e-mail: frankvanschalkwijk@gmail.com; randolph.helfrich@gmail.com

general anesthesia[27,35]. Moreover, the spectral exponent follows a hierarchical organization with overall higher values in early sensorimotor areas as compared to the association cortex[36]. Critically, 1/f noise exhibits spectral power at all frequencies, which raises the intriguing possibility that region-, state-, and task-specific changes in ripple rate largely reflect 1/f noise characteristics. Here, we adopt the term '1/f noise' to highlight its relationship with well-known noise definitions (such as white, pink, and brown noise that correspond to $\chi = 0$, $-1$, and $-2$, respectively) and its detrimental impact on ripple detection.

To address this hypothesis, we combined three experiments conducted during sleep and wakefulness with noise simulations to test if, and to which extent, noise characteristics impact ripple detection. Using simulated 1/f noise with varying spectral exponents, we tested whether ripple detection is modulated as a function of the underlying 1/f noise. In the next step, using experimental data, we specifically tested if more ripples are evident in cortical regions that exhibit a higher spectral exponent (e.g., hippocampus and the medial temporal lobe; MTL) as compared to neocortical association areas (e.g., lateral prefrontal cortex; LPFC). In addition, we determined the relationship of 1/f noise and ripples during different cortical states (i.e., sleep, quiet wakefulness, and cognitive engagement). Given prominent, state-dependent $1/f$ differences wherein the exponents are high during cognitive engagement and lower during sleep, we predicted a higher ripple rate in the wake state as a result of increased $1/f$ noise. To test if state-specific changes in ripple detection reflect an altered 1/f noise characteristic, we conducted analyses on electrodes in the early visual and motor cortex – regions for which no strong prior evidence of ripple occurrence exists. We reasoned that ripple detections should be driven by the previously documented change in 1/f noise[37,38].

## Results

### 1/f noise characteristics drive ripple event detections

Detection of putative ripple events relies on predefined sets of criteria that include band-pass frequency, duration, and amplitude. Numerous detection algorithms have been proposed wherein these criteria are variably applied. Here, we implemented five different algorithms that reflect commonly reported methodological approaches (Table 1). As we do not wish to highlight any specific algorithm or study in particular, we employed adaptions that aim to illustrate the range of possible algorithmic criteria and their impact on ripple detection in various noise scenarios. It is critical to highlight that most algorithms have not been benchmarked in the past; hence, their false positive rate remains unknown. To illustrate this issue, we first detected ripples on real data recorded during the longest consecutive slow-wave sleep period (SWS; duration = 23.5 min) as well as simulated data that matched the spectral exponent of the recording per 30 s segment ($\chi = -2.01 \pm .03$; mean ± SE; $N_{segments} = 47$) for an exemplary single subject and channel (Fig. 1a). Both yielded physiologically plausible

putative ripple events that satisfied commonly employed criteria, including the typical morphology in the time domain (Fig. 1b Left) and the isolated, short-lasted power increase in the ripple-band (80–120 Hz) in time-frequency representations (Fig. 1b Right) without concomitant broadband power changes. Given the short and transient nature of ripples, spectral representations of SWS typically do not exhibit a clear ripple peak, similar to pink or brown noise (Fig. 1c). However, it is critical to highlight that while noise signals are devoid of oscillatory activity, they still contain non-oscillatory, broadband spectral power in the ripple band (Fig. 1c). After identifying candidate events with a detection algorithm, the events will necessarily feature the selection characteristics—which in the case of ripple detection constitutes a clear and distinct peak in the ripple band (red trace in Fig. 1c).

To quantify the relationship between $1/f$ noise and ripple detections, we simulated multiple one-hour-long surrogate EEG traces that followed a $f^\chi$ relationship with varying exponents $\chi$, ranging from $-4.0$ to 0 in 0.1 steps. Note that the surrogate datasets (1/f noise time series) were simulated solely based on the pre-defined spectral exponent. Hence, we did not simulate ripple oscillations on top of the 1/f noise, indicating that all detections were solely obtained from the noise time series, devoid of any true oscillatory activity. We observed that the density of detected ripples (defined as the number of events per second; Hz) differed between algorithms, but always scaled with the spectral exponent (Fig. 1d, all $p < 0.0001$, RM-ANOVA; effect sizes: detector (1): $\eta^2 = 0.99$; (2): $\eta^2 = 0.96$; (3): $\eta^2 = 0.08$; (4): $\eta^2 = 0.95$; (5): $\eta^2 = 0.99$). Critically, all algorithms yielded physiologically plausible ripple waveform shapes as commonly reported (Fig. 1e). However, the precise relationship between spectral exponent and ripple density differed between the detection algorithms: Detector 1 and 2 exhibited an inverted u-shape curve (a direct result of the min. and max. amplitude criteria) where ripple events peaked for exponents of $-1.6$ (detector 1) and $-1.8$ (detector 2). Changing the amplitude cut-off introduced a ceiling effect for higher exponents that approached white noise ($\chi = 0$; detector 5). In contrast, detectors 3 and 4 exhibited an inverse relationship where negative exponents gave rise to more ripple detections. It is noteworthy that this negative relationship (and the overall higher number of detected events) resulted from their duration criteria (cf. Table 1). In sum, this set of findings demonstrates that ripple detection is modulated as a function of the underlying 1/f noise.

### Spectral characteristics of aperiodic activity drive ripple detection during sleep

Having demonstrated that ripple detection depends on 1/f noise characteristics, we next determined how physiologic differences in 1/f activity shape ripple expression. In order to capture the region- and state-specific impact of 1/f noise on putative ripples, we assessed MTL and LPFC activity during wakefulness and SWS—conditions where clear 1/f differences have recently been described. Specifically, the spectral

## Table 1 | Ripple detection algorithms

| Detector | Filtering | Amplitude | Amplitude processing | Amplitude threshold | Duration | Merger | Sleep stage | Artifact rejection (± ripple peak) |
|---|---|---|---|---|---|---|---|---|
| 1 | 80–120 Hz (Butterworth) | Hilbert | 10 Hz low-pass & z-normalization | 2–4 SD Max amp. diff. <2. | 25–200 ms | ± 500 ms | NREM 2–3 | ± 2.5 s |
| 2 | 80–100 Hz (FIR) | R.M.S. (20 ms moving avg.) | NA | 99% of RMS | > 38 ms | NA | NREM 2–4 | ± 1.5 s |
| 3 | 80–120 Hz (Butterworth) | Hilbert | NA | 2–3 SD | > 25 ms | ± 15 ms | NA | NA |
| 4 | 70–180 Hz (FIR) | Hilbert | Extreme values clipped to 4 SD | > 4 SD | 20–200 ms | ± 30 ms | NA | ± 50 ms |
| 5 | 80–120 Hz (Butterworth) | Hilbert | 10 Hz low-pass & z-normalization | 2–4 SD | 20–200 ms | ± 100 ms | NREM 2–3 | ± 1 s |

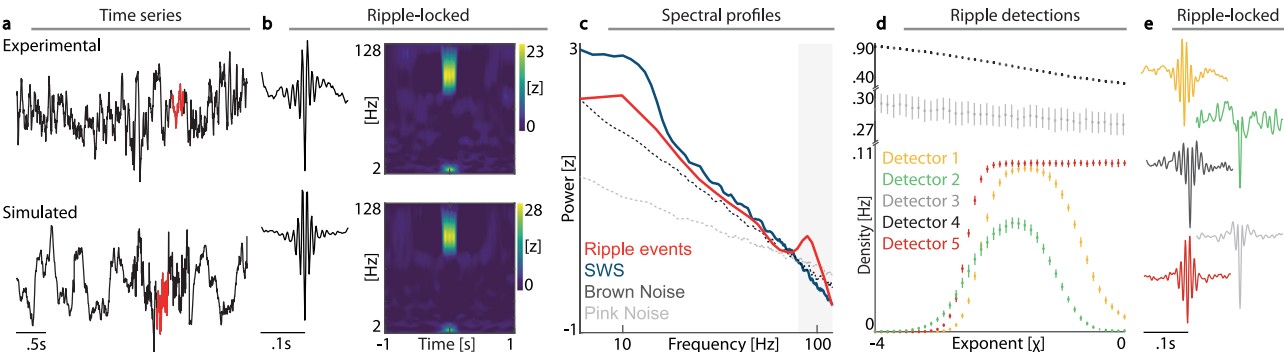

**Fig. 1 | Detecting ripple oscillations in 1/f noise. a** Experimental (Top) and simulated (Bottom) single-channel EEG activity (black) and candidate ripple events (red). **b** Ripple grand-averages for a single subject (Left) and time-frequency representations (Right). Note the striking visual resemblance of experimental and simulated ripples. **c** Spectral power of slow wave sleep activity (SWS, blue; single subject, single channel example), ripples (red; note the distinct peak in the ripple band (80–120 Hz), gray shaded area), as well as pink ($\chi = -1$; light gray dashed line)

and brown noise ($\chi = -2$; dark gray dashed line). Note that colored noise exhibits substantial power in the ripple band. **d** Ripple density (Hz) as a function of the spectral exponent of the simulated data (mean ± SD; 1 h duration; 100 iterations) for five different ripple detectors (different colors). **e** Grand-average event-locked broadband data per detection algorithm (same color conventions as in panel (**d**)), highlighting that all detectors yielded physiologically plausible waveform shapes of candidate ripple events.

exponent is higher in early sensorimotor areas and the MTL as compared to association areas like the LPFC[39,40]. Moreover, the spectral exponent decreases from wake to SWS from approximately −2 to −3, especially in the MTL, while this relationship might be reversed in other regions, such as the LPFC.

To capture 1/f activity and ripples with high spatiotemporal resolution, we analyzed intracranial EEG recordings from the MTL and LPFC during a whole night of sleep ($N = 14$; 82 MTL bipolar contacts, 339 LPFC contacts) while pharmacoresistant patients underwent presurgical invasive monitoring with implanted depth electrodes. We compared the iEEG recordings during sleep to surrogates by simulating the noise profile throughout the night. Specifically, for every 30 s segment of sleep iEEG, we simulated a corresponding segment with the same spectral exponent per channel. Spectral exponents were determined per 30 s segment using FOOOF ("Methods"). The segments were subsequently concatenated, which yielded a 1/f matched surrogate iEEG trace devoid of any oscillatory activity. We then separately detected ripples on the experimental and surrogate data (Fig. 2a) for MTL and LPFC contacts (Fig. 2b). Since all detection algorithms exhibit their own idiosyncrasies, we conducted all subsequent analyses using the first detector. This detector was chosen for the main analyses as it utilizes several characteristics shared across detectors (c.f., Table 1) and yields a representative detection profile on 1/f noise (c.f., Fig. 1d, Supplementary Fig. 1). Note that qualitatively and quantitatively similar results were observed for the other detectors (SI Appendix). A separate detection of interictal epileptiform discharges (IEDs) was conducted for the experimental datasets ("Methods"), thus discarding any candidate ripple events in close temporal proximity of an IED (±2.5 s); thereby minimizing erroneously considering IEDs as ripples.

Qualitatively similar results can be observed with the other detectors as well, revealing a biased relationship between 1/f noise and ripples, albeit that ripple detections differ slightly between distinct algorithms (cf. Fig. 1d). 1/f activity was estimated from FOOOF models with standard parameters ("Methods"). The overall goodness-of-fit of the aperiodic model was high ($R^2 = .9874 \pm .0008$; median ± SEM), indicating a successful parametrization of the underlying power spectra. This decomposition revealed prominent power differences in the ripple band between wakefulness and SWS (Fig. 2c Top) as well as between MTL and LPFC electrodes (Fig. 2c Bottom) as a result of altered 1/f characteristics. Specifically, the spectral exponent was lower in the LPFC (−2.85 ± .03, mean ± SEM) as compared to the MTL (−2.36 ± .07), irrespective of state (Fig. 2d; $t_{419} = 6.43$, $p < .0001$, two-sample t-test). In addition, spectral exponents during SWS differed from wakefulness in both regions. Statistical quantifications were

conducted at the pseudo-population level using repeated-measures ANOVAs (RM-ANOVA) and were further supported by linear mixed-effect models (LME) that accounted for subjects and electrodes. At a pseudo-population level, the spectral exponent was lower during SWS as compared to wakefulness in the MTL ($F_{1,81} = 18.65$, $p < .0001$, $\eta^2 = .19$, RM-ANOVA), and highly comparable findings were observed when accounting for subjects and electrodes using LME models ($p < .0001$, 95% confidence interval [$CI_{95}$] = [−.33, −.12], $t_{162} = -4.35$, $\beta = -.23$, LME). Notably, this relationship was reversed in the LPFC ($F_{1,338} = 203.68$, $p < .0001$, $\eta^2 = .38$, RM-ANOVA; $p < .0001$, [$CI_{95}$] = [.35, .46], $t_{676} = 14.29$, $\beta = .40$, LME).

To understand the relation between the spectral exponent and ripple detection, we analyzed ripple density as a function of the spectral exponent (from −6.0 to 0, in 0.1 steps). For MTL ripples, we observed a broad distribution across many spectral exponents (Fig. 2e Top Left), whereas ripples detected on 1/f noise-matched surrogates fell within a narrower, but substantially overlapping range of exponents (−2.7 to −.4). In total, 58% of empirically observed ripples fell within this noise range. Qualitatively and quantitatively similar results were evident in the LPFC (Fig. 2e Bottom Left), where 28% of ripples were within the noise floor (exponent range: −2.7 to −.4), as well as for different spectral parametrization approaches (Supplementary Fig. 2).

To quantify these observations, we determined ripple density as a function of region and state for both observed and surrogate data (Fig. 2e Right). For the MTL, we observed a significant decrease in ripple density (Fig. 2e Top Right) from wakefulness to SWS (experimental data: $F_{1,81} = 13.61$, p = .0004, $\eta^2 = .14$, RM-ANOVA; $p = .0003$, [$CI_{95}$] = [−.01, 0], $t_{162} = -3.71$, $\beta = -.01$, LME; simulated data: $F_{1,81} = 32.68$, $p < .0001$, $\eta^2 = .29$, RM-ANOVA; $p < .0001$, [$CI_{95}$] = [−.02, −.01], $t_{162} = -5.75$, $\beta = -.02$, LME). This decrease was most strongly observed for simulated as compared to experimental data ($F_{1,162} = 12.28$, $p = .0006$, $\eta^2 = .07$, RM-ANOVA; $p = .0013$, [$CI_{95}$] = [−.02, −005], $t_{324} = -3.25$, $\beta = -.01$, LME).

For the LPFC, we observed significant changes in event density (Fig. 2e Bottom Right), which decreased from wakefulness to SWS for the experimental data ($F_{1,338} = 8.87$, $p = .0031$, $\eta^2 = .03$, RM-ANOVA; $p = .0030$, [$CI_{95}$] = [−.004, −.001], $t_{676} = -2.98$, $\beta = -.002$, LME), whereas simulated data showed increased event density from wakefulness to SWS ($F_{1,338} = 68.45$, $p < .0001$, $\eta^2 = .17$, RM-ANOVA; $p < .0001$, [$CI_{95}$] = [.01, .01], $t_{676} = 8.29$, $\beta = .01$; LME). We consequently observed an interaction effect wherein conditional changes in ripple density were different between experimental and simulated data ($F_{1,676} = 68.08$, $p < .0001$, $\eta^2 = .09$, RM-ANOVA; $p < .0001$, [$CI_{95}$] = [.01, .01], $t_{1352} = 6.32$, $\beta = .01$, LME).

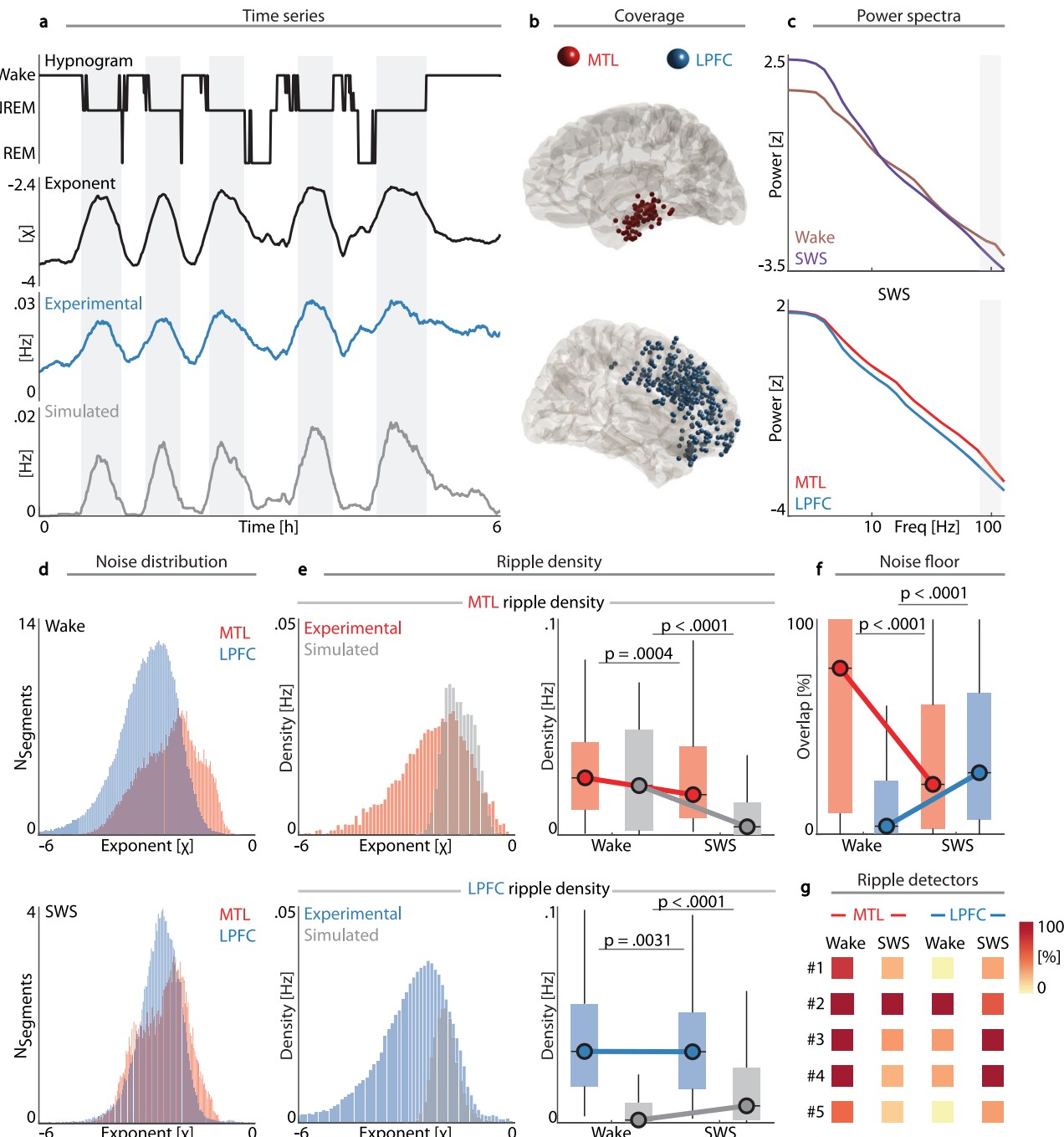

**Fig. 2 | 1/f noise characteristics explain region- and state-specific ripple variability. a** From Top to Bottom: Sleep hypnogram (single subject); time-resolved spectral exponent (black); event density (Hz) of detected ripples (blue) for one prefrontal channel. Note the resemblance of detected ripples on simulated data (gray) that matched the noise characteristics of the experimental data. **b** Group level ($N = 14$) coverage in the medial temporal lobe (MTL; $N_{electrodes} = 82$; Top; red) and lateral prefrontal cortex (LPFC; $N_{electrodes} = 339$; Bottom; blue). **c** State- (Top; single subject, single channel) and region-dependent (Bottom; group level) power differences in the ripple band (shaded area). **d** Top: Distribution of spectral exponents for MTL and LPFC during wakefulness. Bottom: Distribution of spectral exponents during slow-wave sleep (SWS). **e** Region- and state-specific relationship between noise and ripples. Top Left: Distribution of MTL ripple density as a function of the spectral exponent (red). Ripple detection in noise-matched epochs (gray) highlights that MTL ripples in epochs with an exponent > −2.7 are within the noise range. Bottom Left: Distribution of LPFC ripple density relative to the spectral

exponent (blue; same conventions as the top panel and panel (**d**)). Top Right: Mean MTL ripple density per state. We observed more ripples during wakefulness than during SWS (experimental: $F_{1,81} = 13.61$, $p = .0004$; simulated: $F_{1,81} = 32.68$, $p < .0001$; RM-ANOVA) as a result of higher spectral exponents (cf. top panel (**c**)). Boxplots represent median (middle line), $25^{th}$ and $75^{th}$ percentiles (box) and extreme values (whiskers). Bottom Right: Mean LPFC ripple density per state (experimental: $F_{1,338} = 8.87$, $p = .0031$; simulated: $F_{1,338} = 68.45$, $p < .0001$; RM-ANOVA). Same conventions as the left and top panels. **f** Differences in ripple percentage within the noise floor during wake and SWS (MTL: $F_{1,81} = 21.16$, $p < .0001$; PFC: $F_{1,338} = 122.98$, $p < .0001$; RM-ANOVA). Same conventions as panel (**e**). **g** Noise-floor estimates across states and regions for five ripple detectors. The majority of detection algorithms demonstrate the same state and regional differences. Colors depict median values per detector, state, and region (the first row corresponds to values displayed in panel (**f**)).

To illustrate and summarize these results, we calculated the percent overlap of observed ripples relative to the surrogate distribution (Fig. 2f). In the MTL, we observed that the median proportion of wake ripples falling within the noise floor was 77%, while only 23% during SWS could be attributed to the 1/f characteristics. In the LPFC, 4% of ripple detections during wakefulness and 28% during SWS possibly reflected false positives. Statistical quantification confirmed that false positive ripple detections significantly decreased from wakefulness to SWS in the MTL ($F_{1,81} = 21.16$, $p < .0001$, $\eta^2 = .21$, RM-ANOVA; $p < .0001$, $[CI_{95}] = [−37.33, −15.0]$, $t_{162} = −4.63$, $\beta = −26.17$, LME), but showed the reversed pattern in the LPFC ($F_{1,338} = 122.98$, $p < .0001$, $\eta^2 = .27$, RM-ANOVA; $p < .0001$, $[CI_{95}] = [17.7, 25.3]$, $t_{676} = 11.11$, $\beta = 21.54$, LME). We consequently observed a significant interaction effect between the two regions ($F_{1,419} = 97.28$, $p < .0001$, $\eta^2 = .19$, RM-ANOVA; $p < .0001$, $[CI_{95}] = [38.24, 57.18]$, $t_{838} = 9.89$, $\beta = 47.71$, LME). Similar effects can be observed for all other detectors (Fig. 2g, Supplementary Fig. 3). In sum, these results indicate state- and region-dependent modulation of ripple density that follows a $1/f^x$ relationship.

To determine if ripple density was biased by 1/f noise, we employed partial correlations to control for the 1/f noise floor as defined by the surrogate ripple density. This analysis demonstrated that the positive correlation between the spectral exponent and ripple density observed for MTL channels ($r_s = .37$, $p = .006$) can be accounted for when including surrogate ripple density as a confounder ($r = .05$, $p = .6687$). LPFC channels similarly showed that ripple density and spectral exponent values were significantly correlated ($r_s = .62$, $p < .0001$), which was significantly reduced ($p = .0009$, comparison of correlation coefficients) when accounting for the surrogate ripple density (partial correlation: $r = .44$, $p < .0001$).

Collectively, this set of findings demonstrates that region- and state-specific differences in 1/f noise characteristics severely impact ripple detection, especially if the detector noise susceptibility (Fig. 1d) peaks within the experimental condition, as exemplified for MTL ripples. While the spectral exponent of the MTL during wakefulness matched the noise profile, the spectral exponent was lower during SWS and outside of the detector noise range; thus, explaining why more false positive ripples were detected during wakefulness as compared to SWS. Note that different detection algorithms exhibit varying detection patterns on 1/f noise, but give rise to qualitatively comparable results (Supplementary Fig. 1, Supplementary Fig. 3).

## Wake ripples result from a spectral exponent increase during tasks

Modulation of 1/f noise is not limited to the wake-sleep cycle, but also occurs physiologically over the course of single trials during cognitive tasks, such as perception, attention, and memory encoding[29,41]. Given the recent interest in wake ripples, we hypothesized that ripple detections often reflected altered 1/f noise characteristics rather than genuine ripple activity. As we did not intend to single out a specific study or paradigm, we sought to illustrate this issue in early visual and motor cortex – two regions where ripple activity has previously not been reported and where ripples are not typically thought to constitute key elements of their functional architecture. We specifically assessed two datasets where Miller and colleagues previously reported prominent 1/f (broadband) modulations during task engagement as compared to baseline intervals[37,38]. We hypothesized that ripple detection should reflect the 1/f modulation through increased ripple detections during the task epoch. Moreover, we reasoned that if this ripple increase solely stems from a change in 1/f noise, then ripple detection on simulated, noise-matched surrogates should yield a comparable number of ripples. Ripple detection was conducted identically to our analysis on the Sleep dataset, and ripples in close temporal proximity to IEDs (±2.5 s) were discarded.

To demonstrate the feasibility of this approach, we first simulated 1/f noise with varying exponents over several seconds that might

correspond to different task epochs (Fig. 3a, Supplementary Fig. 1). As expected, ripple detection was systematically biased to the state with the highest spectral exponent. This bias was driven by the predefined amplitude threshold criterion in the ripple detection algorithm. This fixed threshold is derived from the mean and variance of the band-pass-filtered signal envelope across the entire recording; a methodological approach that is common to most detection algorithms (cf. Table 1).

Next, we analyzed intracranial recordings from subdural electrodes placed over occipital areas (Fig. 3b; 34 electrodes in 5 patients) while patients searched for a visual target[37]. In this task, patients viewed a 5 × 4 colored grid wherein a white star indicated the target stimulus and a black arrow indicated the target direction (down, up, left, or right). Patients had to verbally report the color of the adjacent grid coordinate next to the target stimulus in the cued direction. Ripple detection yielded a plausible grand-average waveform (Fig. 3c) and, as hypothesized, both the spectral exponent (Fig. 3d Top; Supplementary Fig. 4a) and ripple density (Fig. 3d Center) increased during the task epochs as compared to the inter-trial interval (ITI). The overall goodness-of-fit of the FOOOF model to the power spectra was high ($R^2 = .8793 ± .0059$; median ± SEM). We again simulated surrogate trials that matched the 1/f noise exponents during the ITI and task epochs (as outlined in Fig. 3a; for 1 s segments without overlap). Ripple detection on the surrogate data also increased during the simulated task epoch (Fig. 3d Bottom), indicative of the fact that ripple detection reflected variations in the 1/f noise.

Similar to analyses on our sleep dataset, statistical quantifications were conducted at the pseudo-population level using RM-ANOVAs and were supported by LMEs that accounted for subjects and electrodes. These analyses demonstrated a significant effect of condition, leading to increases during the task epoch for the spectral exponent (Fig. 3e Top; $F_{2,66} = 13.58$, $p = .0008$, $\eta^2 = .29$, RM-ANOVA; $ITI_1$: $p = .0003$, $[CI_{95}] = [−.58, −.18]$, $t_{99} = −3.78$, $\beta = −.38$, LME; $ITI_2$: $p = .0004$, $[CI_{95}] = [−.57, −.17]$, $t_{99} = −3.70$, $\beta = −.37$, LME), observed ripple density (Fig. 3e Bottom; $F_{2,66} = 59.77$, $p < .0001$, $\eta^2 = .64$, RM-ANOVA; $ITI_1$: $p < .0001$, $[CI_{95}] = [−.09, −.05]$, $t_{99} = −7.88$, $\beta = −.07$, LME; $ITI_2$: $p < .0001$, $[CI_{95}] = [−.09, −.05]$, $t_{99} = −7.82$, $\beta = −.07$, LME), and surrogate ripple density ($F_{2,66} = 23.75$, $p < .0001$, $\eta^2 = .42$, RM-ANOVA; $ITI_1$: $p < .0001$, $[CI_{95}] = [−.10, −.04]$, $t_{99} = −5.06$, $\beta = −.07$, LME; $ITI_2$: $p < .0001$, $[CI_{95}] = [−.10, −.04]$, $t_{99} = −4.84$, $\beta = −.07$; LME). Supplementary analyses demonstrate that modulation of the spectral exponent by task is irrespective of the frequency range used for spectral parameterization (Supplementary Fig. 5a, b).

To determine if the increase in ripple density during task execution was only driven by the relationship to 1/f noise ($r_s = .47$, $p = .0047$), we employed partial correlations to control for the 1/f noise floor as defined by the surrogate ripple density. This analysis yielded a non-significant correlation ($r = −.16$, $p = .3730$), indicating that a change in ripple density can be explained by the spectral dynamics of 1/f noise.

In order to demonstrate that this relationship is not exclusive to the visual cortex, we sought to replicate this set of findings in a second dataset in yet another region where ripples typically do not occur physiologically. We analyzed intracranial EEG data recorded from the motor cortex (Fig. 3f; 99 electrodes across 18 patients) while patients were instructed to execute simple, repetitive movements of the hand (i.e., clenching and releasing a fist) or tongue (protrusion and retraction of the tongue) at approximately 1–2 Hz for a duration of 3 s[38]. After an ITI, patients received a visual cue that indicated the respective body movement. All analytical steps matched the Visual Search Task (Fig. 3b–e). Again, the overall goodness-of-fit of the FOOOF model to the power spectra was high ($R^2 = .9428 ± .0026$; median ± SEM). We observed a qualitatively and quantitatively highly comparable set of findings as compared to the previously discussed Visual Search Task. Ripple detection yielded a plausible grand-average waveform (Fig. 3g) and significant increases were observed during the task epoch for the spectral exponent ($F_{2,196} = 64.27$, $p < .0001$, $\eta^2 = .40$, RM-ANOVA; $ITI_1$:

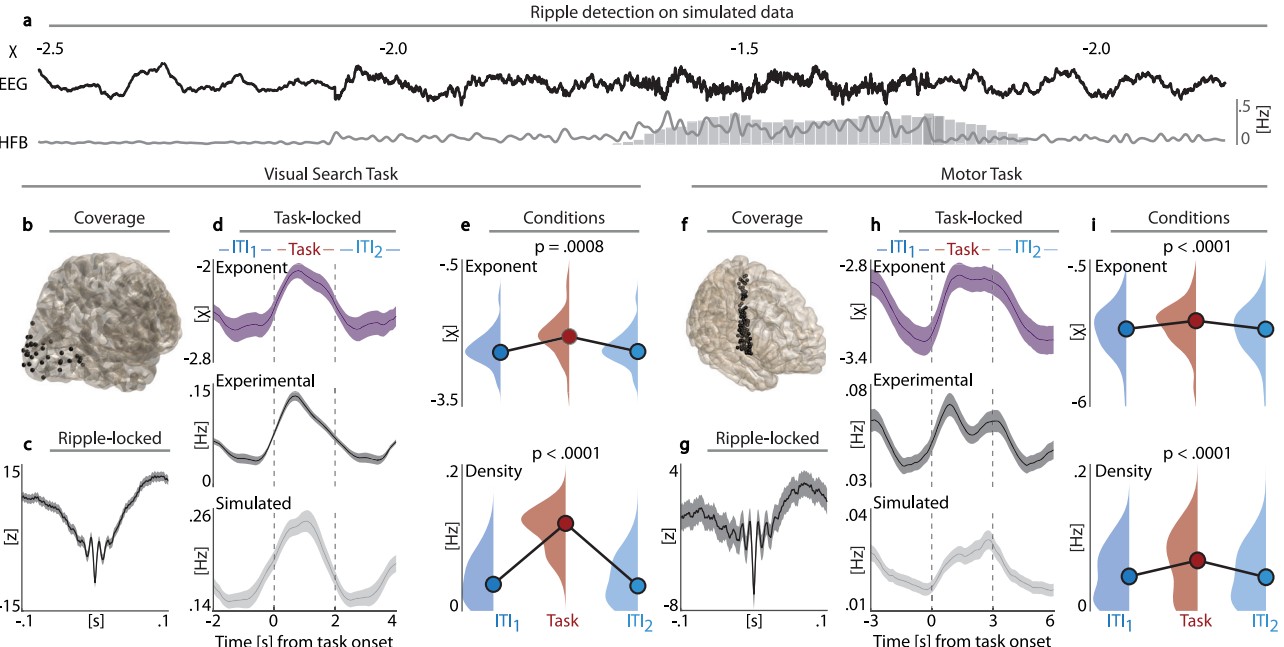

**Fig. 3 | Stimulus-locked ripples reflect changes in 1/f noise. a** Illustration of how ripple detection scales as a function of the 1/f exponent. Upper row: Simulated signal with varying exponents (black) in 3 s segments. Lower row: Ripple-band activity (gray line; 80–120 Hz; z-normalized). Note that ripple occurrence (gray bars; 100 iterations) occurred during the highest spectral exponent, thus revealing a systematic bias in ripple detections. **b–e** Noise and ripple modulations during visual search. **b** Occipital electrode coverage (N = 5; $N_{electrodes}$ = 34). **c** Ripple grand average during visual search in occipital cortex (mean ± SEM). **d** Upper row: Time-resolved spectral exponent (purple; mean ± SEM). Note the prominent increase during task execution as compared to the pre and post-inter-trial intervals (ITI). Middle row: Ripple distribution over the course of the trial, again, with a prominent increase during the task period. Bottom row: Ripple distribution on surrogate data

that matched the 1/f noise characteristics of the original recording and closely matched the experimental condition. **e** Statistical quantification. The spectral exponent (Top; $F_{2,66}$ = 13.58, p = .0008; RM-ANOVA) and ripple density (Bottom; $F_{2,66}$ = 59.77, p < .0001; RM-ANOVA) increased during the task epoch as compared to the ITI. Correlational analyses demonstrated that the positive relation between ripple density and spectral exponent ($r_s$ = .47, p = .0047; Spearman correlation) can be explained by the spectral dynamics of 1/f noise (r = −.16, p = .3730; partial correlation). **f–i** Noise and ripple modulations during movement execution as recorded from motor cortex (N = 18; $N_{electrodes}$ = 99) replicate the increase of the spectral exponent ($F_{2,196}$ = 64.27, p < .0001; RM-ANOVA) and ripple density ($F_{2,196}$ = 21.33, p < .0001; RM-ANOVA) during the task epoch as compared to the ITI (same conventions as in panels (**b**–**d**)).

p < .0001, [$CI_{95}$] = [−.24, −.15], $t_{294}$ = −8.07, β = −.20, LME; $ITI_2$: p < .0001, [$CI_{95}$] = [−.25, −.15], $t_{294}$ = −8.12, β = −.20; LME), observed ripple density ($F_{2,196}$ = 21.33, p < .0001, $η^2$ = .18, RM-ANOVA; $ITI_1$: p < .0001, [$CI_{95}$] = [−.02, −.01], $t_{294}$ = −4.64, β = −.01, LME; $ITI_2$: p < .0001, [$CI_{95}$] = [−.02, −.01], $t_{294}$ = −4.69, β = −.01; LME), and surrogate ripple density ($F_{2,196}$ = 10.94, p = .0012, $η^2$ = .10, RM-ANOVA; $ITI_1$: p = .0012, [$CI_{95}$] = [−.0101, −.0025], $t_{294}$ = −3.27, β = .−.01, LME; $ITI_2$: p = .0007, [$CI_{95}$] = [−.0102, −.0027], $t_{294}$ = −3.41, β = −.01; LME) (Fig. 3h, i, Supplementary Fig. 4b). Additional analyses demonstrate that modulation of the spectral exponent by task is irrespective of the frequency range used for spectral parameterization (Supplementary Fig. 5c, d). Correlational analyses again demonstrate that ripple density and spectral exponent values were significantly correlated ($r_s$ = .05, p = .0264), which resulted in a non-significant partial correlation when controlling for surrogate ripple density as a confounding variable (r = .01, p = .2934).

In sum, these results clearly demonstrate that ripples during wakefulness and cognitive engagement largely reflect false positive detections within 1/f noise and not genuine ripple oscillations. We purposefully decided to illustrate this relationship in two cortical areas and task conditions that have previously not been associated with ripples. However, since many cognitive tasks, such as associative memory tasks, rely on visual input and overt motor responses, the present results can be extrapolated to these related contexts. While several papers reported the presence of wake ripples during memory tasks[10,14,16–18], a largely unrelated line of inquiry focusing on aperiodic activity reported modulations of the spectral exponent during cognitive engagement. Our findings now offer a more parsimonious explanation for why ripple density increases during task epochs.

## Discussion

Hippocampal ripples play a crucial role in sleep, memory, and cognition. While their morphology, underlying cellular mechanisms, and primary functions are well characterized in the rodent hippocampal CA1 region[1], it remains unclear whether ripples observed during wakefulness or in the human neocortex are governed by similar principles. Recently, putative ripples have been described as a wide-spread neocortical phenomena in a range of behavioral contexts in the wake state[10,13,14,16–18]. In simulations and intracranial human recordings across three experiments, we demonstrate that putative ripples largely result from different 1/f noise characteristics that systematically vary as a function of cortical region, state, and task demands. It is critical to highlight that all detection algorithms identified ripples for simulated datasets that solely consisted of colored noise with pre-defined spectral exponents. Our results provide a framework based on surrogate 1/f noise simulations to infer a state-specific false positive rate. Moreover, these findings reveal the noise characteristics of various detection algorithms and highlight the need to account for context-dependent 1/f activity during ripple detection.

Neural oscillations, such as ripples, constitute discrete events that can be detected by eye in the time domain[1]. In the rodent brain, there is a long-standing interest in slow oscillations, theta, and ripples, whereas alpha, beta, slow oscillations, and sleep spindles dominate the human literature[42,43]. Recently, it became evident that 1/f broadband activity may conflate the estimation of oscillatory signatures; thus, necessitating a separation through spectral parametrization[36]. An emerging line of research demonstrates that variations in 1/f background activity track cognitive processes more closely than presumed oscillatory signatures in various behavioral contexts[29,30,44]. Likewise, it had been

demonstrated that 1/f activity clearly distinguishes different sleep stages[33,35,45]. Critically, ripples have been primarily described during NREM sleep and wakefulness, cognitive states characterized by spectral exponents ranging from −2 to −3. We now provide an explanation why most ripples are being detected during these cognitive states, as our simulations establish that the noise sensitivity of commonly employed ripple detectors (e.g., 1 and 2) peaks within this range (cf. Fig. 1d). In contrast, ripples are typically not described during REM sleep, which exhibits a spectral exponent closer to −4, hence, outside of the detectors' noise floors. In line with this consideration, it has been shown that replay events in the human brain are often accompanied by broadband power increases[25] that do not necessarily reflect narrow-band ripple oscillations.

In rodents, hippocampal SWRs are well characterized in terms of their generative mechanisms and functional roles. However, a key unresolved question is whether ripples similarly contribute to human cognition. To date, direct comparative evidence remains sparse[15,22,24,46]. The main similarity between rodent SWR and hippocampal ripples in humans is their oscillatory coupling to other sleep oscillations, such as spindles and slow waves[9,11,12,47]. Critically, ripple detections during SWS have a much lower false positive rate than during wakefulness as a direct consequence of the lower spectral exponent during SWS (cf. Fig. 2e, f; but note the dependence on the specific detector, cf. Fig. 1d). This is in line with the observation that neural firing is synchronized to putative ripples in the human hippocampus[48], thus mirroring observations in the rodent brain[49]. It is conceivable that especially neocortical ripple detections, as recently described in rodent[50] and human posterior parietal cortex (PPC)[13,14], are at least partially attributable to altered noise characteristics[36]. It is intriguing that converging evidence across both species demonstrated different functional consequences of hippocampal and neocortical ripples. While hippocampal ripples were shown to engage the hippocampal-neocortical dialog[15,50–53], neocortical ripples (esp. PFC) decreased inter-areal information flow to shield neocortical information recapitulation from interfering hippocampal input[15,53]. Given the 1/f noise characteristics of SWS and the PFC, it is unlikely that the observed effects can be solely attributed to false positive detections. Future comparative studies will need to determine the extent to which high-frequency phenomena correspond to SWR in CA1. In the meantime, ripple signatures remain valuable markers of discrete events reflecting large-scale network engagement that predict behavior[10,14,16], provided that noise characteristics are well defined[20].

The gold standard for SWR detection in rodents is high-density laminar recordings that enable the visual identification of the distinct sharp-wave component arising in CA3 and the ripple component in CA1[1]. Hence, most detection algorithms that are employed in rodents take advantage of detecting multiple waveform characteristics[54,55]. As virtually all intracranial EEG recordings in the human brain are obtained from epileptic patients, it is not feasible to detect sharp wave components, as these are also hallmark interictal discharges. Hence, it proves challenging to detect ripple events based on visual inspection, as even experts often cannot reach consensus[24]. To overcome these challenges, it is critical to move towards a consensus for the recording, detection, and reporting of ripples[20], for which the authors provided a series of recommendations. An important step is the comparison to ripples detected during NREM sleep. The present study demonstrates a challenging aspect of this control analysis, as even simulated data may contain physiologically plausible ripple events that resemble the typical morphology in the time and frequency domain (Fig. 1). A second recommendation is accounting for EEG background activity. We demonstrate how background noise characteristics differ between states as well as regions (Fig. 2), which has not been accounted for in previously employed detection approaches. Crucially, changes in background activity may also give rise to spurious ripple detections in the neocortex (Fig. 3). Such false positive ripple detections result from fixed amplitude and duration thresholds applied to the signal envelope of the band-pass filtered EEG recordings – a methodology utilized by many detection algorithms (c.f. Table 1). These amplitude thresholds are often based on the standard deviation of the background activity across the entire recording, thus not accounting for possible differences between brain states. We now demonstrate how noise floors can be estimated per state and region, and offer an additional approach to establish a false-positive detection rate. Note that the choice of frequency band, ripple duration, and various other detection criteria constitute additional sources of detection bias that likely account for the variability across studies. Further recommendations, such as simultaneous recording of micro- and macro-LFP in the hippocampus, localization to hippocampal subfields and layers, and investigating the effect of anti-seizure medication on ripple occurrence and sleep, provide avenues for future studies to demonstrate how ripple detection can be improved.

So how can we improve the detection of ripple events? In addition to established practices as discarding ripples close to epileptic discharges[12] or merging events that occur within a narrow time window[16], our approach now outlines a simulation-based approach to estimate the noise floor that provides a baseline of what can be expected in a given context or state. However, it is conceivable that some detections within the noise floor reflect true ripples. So how can we separate spurious from correct event detections? One option is to record simultaneous single-unit activity through microwire[18,48] or microelectrode arrays[56], which, however, is not as widely available as clinical macro-electrode recordings. In addition to comparing putative wake ripples to NREM ripples[20], another possible solution is considering secondary criteria, such as oscillatory coupling to slow waves and spindles, which is only feasible during non-REM sleep. Other secondary criteria might include ripple co-occurrence on multiple channels[50], determining ripple-related consequences, such as memory reactivations[17] and replay[26,57], or contrasting putative ripple events with surrogate event detections that match spectral, but not waveform features.

In this study, we evaluated five different ripple detectors. These detectors, or close variants thereof, have been used to study hippocampal[7,12,51,58] as well as neocortical[13–16,56] ripples during NREM sleep and wakefulness. In addition, hippocampal ripples were reported during memory encoding and retrieval[10,17,18,59], wherein differences in ripple rate between remembered and forgotten items during encoding or retrieval provide strong behavioral evidence. In light of the here-reported association of 1/f noise and ripple detection, it might be conceivable that ripples, as previously reported, mainly index altered excitability dynamics and local cortical activation. This is in accordance with a large body of literature that shows that increased high-frequency band activity (sometimes termed broadband or high-gamma) precisely indexes local cortical activation[60–62]. One conceivable approach for future studies is to closely examine oscillatory signatures (such as ripples) as well as aperiodic markers to infer to which extent these two markers overlap or reflect dissociable signatures. In the context of memory formation, some evidence suggests that these reflect dissociable signatures[63].

Although our study focused on ripples, it is conceivable that similar considerations apply to the detection of other discrete, short-lasted phenomena in electrophysiological recordings. For example, detection of beta or gamma bursts[64], gamma oscillations[65,66], high-frequency oscillations (HFOs)[67,68], or fast ripples (pathologic ripples)[21,69] might equally be affected by altered 1/f noise. In the case of epilepsy, it has been suggested that HFOs might delineate the seizure onset zone (SOZ)[21]. However, recent evidence revealed higher spectral exponents and flatter power spectra in epilepsy[70–73]; hence, giving rise to the viable hypothesis that HFOs may result from an increase of the spectral exponent as a result of the hyperexcitability syndrome. Our results demonstrate increased ripple rates as a result of an increased spectral exponent during task performance (Fig. 3). It is of great interest to assess if oscillatory beta/gamma bursts during cognitive

processing also reflect altered 1/f characteristics, and to determine if bursts that often vary substantially in peak frequency constitute genuine network oscillations. In sum, these considerations provide a more parsimonious explanation for why many oscillatory phenomena in the human brain are transient and short-lived, often with substantial variability of their peak frequency.

In sum, our results demonstrate that up to 77% of human ripple events during wakefulness result from altered region- or state-dependent 1/f noise characteristics. We introduced a simulation-based approach to estimate the false positive rate and systematically evaluated each detection algorithm's susceptibility to 1/f noise, demonstrating that this factor is a major source of the variability reported across studies. Our results posit that ripple algorithms need to be benchmarked against the noise floor. This involves estimating spectral exponents, generating matched colored noise, applying identical detection methods to both real and simulated data, and quantifying false positives within the noise floor. As with ripple detection, there are multiple ways to define aperiodic 1/f noise activity. Despite methodological differences, most studies consistently report spectral exponents ranging from approximately −1 to −5. This suggests a shared underlying spectral structure, albeit the fact that specific details may vary depending on how noise is modeled. In our simulations, we used time series characterized by a single exponent across the recording, or characterized by dynamic exponents across segments. In contrast, real neural recordings often exhibit multiscale dynamics[40,74], reflecting multiple overlapping 1/f components. This complexity poses significant challenges for the reliable detection of discrete events. Hence, an unmet need is the development of robust criteria for identifying individual events—potentially leveraging artificial intelligence, machine learning, and simulation-based inference methods that take their spectral, temporal, broadband, oscillatory, and coupling features into account.

## Methods
### Participants
**Study 1 (sleep).** Sleep recordings were conducted in patients with pharmacoresistant epilepsy ($N = 14$; $M_{age} = 36.79 \pm 13.28$ years, mean ± SD, range 19–58 years; 9 female) undergoing pre-surgical invasive monitoring to delineate the seizure onset zone. Intracranial depth electrodes (Ad-Tech) were implanted in all subjects, and two patients received additional subdural grid electrodes. Electrodes were localized in native space by two independent neurologists, and group-level topographical visualization was done in normalized MNI space using Matlab R2021a (MathWorks Inc.) and Fieldtrip (fieldtrip-20241025)[75] (ft_plot_mesh) (Fig. 2b). Specific localization was based on co-registered pre- and post-implantation MRIs as well as a high-resolution anatomical atlas as previously outlined[11,76–78]. In line with our recent studies[9,15], only electrodes within the MTL (CA1, CA3/dentate gyrus, subiculum, entorhinal cortex, perirhinal cortex, parahippocampal gyrus, and temporal pole; $N = 82$) or LPFC (medial and lateral prefrontal, as well as orbitofrontal cortices; $N = 339$) were considered for analyses. Electrodes localized to the seizure onset zone were excluded. All recordings were conducted at the University of California Irvine Medical Center, USA. Informed consent was obtained from all subjects prior to study participation. Data acquisition and analyses protocols were approved by the Institutional Review Board at the University of California, Irvine (protocol number: 2014–1522) and the Committee for Protection of Human Subjects at the University of California, Berkeley (Protocol number: 2010–02–783). The study was conducted in accordance with the 6th Declaration of Helsinki.

**Study 2 (visual search task).** Data were recorded from epilepsy patients ($N = 5$) as part of their clinical monitoring to delineate the seizure onset zone. Intracranial platinum subdural grid and strip electrodes (Ad-Tech) were placed at frontal, parietal, temporal, and occipital cortex. Recordings were sampled at 1000 Hz, referenced to scalp and ground electrodes, and band-pass filtered (0.15–200 Hz). Analyses only included electrodes localized in the visual cortex. Group-level topographical visualization was done in normalized MNI space using Matlab and Fieldtrip (ft_plot_mesh) (Fig. 3b; $N_{electrodes} = 34$; VTPM atlas)[79]. All patients participated in a purely voluntary manner, after providing informed written consent, under experimental protocols approved by the Institutional Review Board of the University of Washington (#12193). All patient data was anonymized according to IRB protocol, in accordance with HIPAA mandate. These data originally appeared in the manuscript titled Dynamic modulation of local population activity by rhythm phase in human occipital cortex during a visual search task published in Frontiers in Human Neuroscience in 2010[37].

**Study 3 (motor task).** Data were recorded from epileptic patients ($N = 19$) as part of their clinical monitoring to delineate the seizure onset zone. One subject was excluded from analyses because of a shorter trial length (2 s instead of 3 s duration). Intracranial platinum subdural grid and strip electrodes (Ad-Tech) were placed independently following clinical criteria. Recordings were sampled at 1000 Hz, referenced to scalp and ground electrodes, and band-pass filtered (0.15 or 0.3–200 Hz). Analyses only included electrodes localized in the motor cortex. Group-level topographical visualization was done in normalized MNI space using Matlab and Fieldtrip (ft_plot_mesh) (Fig. 3f; $N_{electrodes} = 99$; Brainnetome atlas [area 4])[80]. All patients participated in a purely voluntary manner, after providing informed written consent, under experimental protocols approved by the Institutional Review Board of the University of Washington (#12193). All patient data was anonymized according to IRB protocol, in accordance with HIPAA mandate. It was made available through the library described in A Library of Human Electrocorticographic Data and Analyses by Kai Miller[81], freely available at https://searchworks. stanford.edu/view/zk881ps0522. All patient data was anonymized according to IRB protocol, in accordance with HIPAA mandate. These data originally appeared in the manuscript titled Spectral Changes in Cortical Surface Potentials during Motor Movement published in Journal of Neuroscience in 2007[38].

### Procedures
**Study 1 (sleep monitoring).** EEG data were simultaneously recorded from intracranial electrodes located in the MTL and LPFC, as well as scalp electrodes (Fz, Cz, C3, C4, and Oz; localized in accordance with the international 10–20 system). Additional electrooculography (EOG; four electrodes placed on the left and right outer canthi) and electromyography (EMG) data were recorded. Data were recorded using a Nihon Kohden recording system (model JE120A; 256-channels), and were digitally sampled at 5000 Hz and analog filtered at .01 Hz. A seizure-free full-night recording was analyzed per patient, with recordings starting between 19:30-22:00 and lasting for 10–12 h. An expert rater scored all recordings according to established criteria (Fig. 2a Top)[82]. Average sleep duration was $447 \pm 123$ min (mean ± SD; range 181–649 min), of which 13% ± 9% was spent in SWS (range 2–36%). Further details regarding sleep recordings are described in detail in our previous study[15].

**Study 2 (visual search task).** Task details are described in detail by Miller et al[37]. In short, the Visual Search Task presented subjects with numerous stimuli (2 s) separated by Inter-Trial-Intervals (ITIs; 2 s) that showed subjects a blank screen using a LCD monitor at 1 m distance. Each stimulus showed (1) a 5 × 4 grid of colored squares (1 × 1 cm), (2) a white star located in the center of a random colored box, and (3) an arrow indicating a direction (up, down, left, & right). Subjects were instructed to verbally report the color of the box adjacent to the star following the arrow's direction.

**Study 3 (motor task).** Task details are described in detail by Miller et al[38]. In short, patients were required to perform simple, repetitive motor tasks of the hand (flexion and extension of all fingers) or tongue (opening of mouth with protrusion and retraction of the tongue) at ~1–2 Hz (i.e., repetitive motion rather than tonic contraction). Trials (3 s) were separated by inter-trial-intervals (ITIs; 3 s) during which subjects were at rest and presented with a blank screen. Movement was performed contralateral to the side of cortical grid placement. Trial on- and offset were visually indicated by the presence or absence of a written word indicating the type of movement within a $10 \times 10$ cm presentation window at a distance of 75–100 cm from the patient.

## Data processing

Intracranial EEG from the Sleep dataset were demeaned, detrended, rereferenced to a bipolar montage, and downsampled to 500 Hz. Simultaneously recorded scalp EEG data utilized for sleep scoring were demeaned, detrended, referenced to the common average over all scalp electrodes, low-pass filtered (50 Hz), and downsampled to 500 Hz. Data from the Visual Search Task and Motor Task were rereferenced to the common average. All data preprocessing and analyses were conducted using Matlab R2021a (MathWorks Inc.). Data preprocessing, filtering, and segmentation were conducted using Fieldtrip (fieldtrip-20241025)[75] and EEGlab (eeglab2020_0)[83] in addition to custom code.

## Simulations

Colored noise were simulated using the colored_noise function[84]. Here, white noise is filtered to create colored noise characterized with a defined spectral exponent of its power spectrum density. Data were iteratively simulated ($N = 100$) for a range of spectral exponents (−4 to 0 in .1 steps) at 1000 Hz, followed by minimum standard EEG preprocessing steps where the data were demeaned, detrended, band-pass filtered (.5–150 Hz), and z-normalized. Data were simulated with a constant spectral exponent (Fig. 1a) or approximated the spectral exponent of the recorded iEEG data per segment over time (Fig. 2A: 30 s segments without overlap; Fig. 3a, d, h; 1 s segments without overlap). Simulated data were created per segment per channel and subsequently concatenated. It is critical to highlight that no simulated ripples were added to the surrogate datasets.

## Event detections

Candidate ripple events were detected on the continuous data using five detection algorithms inspired by recently published algorithms that were utilized in human intracranial studies (Table 1).

**Detector 1.** Data were band-pass filtered (80–120 Hz; Butterworth), Hilbert-transformed to extract the absolute instantaneous amplitude, smoothed (10 Hz low-pass filter; Butterworth), and z-normalized. Candidate ripple events were detected when the Hilbert envelope exceeded the threshold of 2 SD above the mean for a duration of 25–200 ms. Candidate events with an amplitude between 2–4 SD, frequency > 80 Hz, peak-to-peak amplitude differential <2, and no overlap with IEDs ± 2.5 s relative to the ripple trough were considered for analysis. Events were aligned to the nearest trough in the broadband data.

**Detector 2.** Data were high-pass filtered (80 Hz; FIR, filter order 3) and subsequently low-pass filtered (100 Hz; FIR, filter order 3). The absolute signal envelope was extracted using the Hilbert transform, smoothed (20 ms smoothing), and z-normalized. Candidate events were detected when the signal envelope exceeded the 99% of R.M.S. for a minimum duration of 38 ms (i.e., minimum 3 cycles at 80 Hz) and maximum of 100 ms. duration. Candidate events were aligned to the nearest trough in the broadband data, and those within temporal proximity of IEDs (± 1.5 s) were rejected.

**Detector 3.** Data were band-pass filtered (80–120 Hz; Butterworth; two-pass, filter order 2), after which the absolute signal envelope was extracted using the Hilbert transform. Candidate events were detected when the signal envelope exceeded 2 SD above the mean for a minimum of 25 ms. Events ± 15 ms were merged. Events were aligned to the nearest trough in the broadband data.

**Detector 4.** Data were band-pass filtered (70–180 Hz; FIR; two-pass, filter order 5). Signal envelope was extracted using the Hilbert transform. Extreme values (> 4 SD) were clipped to 4 SD, after which the clipped envelope was squared and smoothed (40 Hz low-pass Kaiser FIR filter). Mean and SD of the transformed data were utilized for subsequent event detection on the original (squared but unclipped) signal envelope. Candidate events were detected when the squared signal envelope exceeded 4 SD above the mean for a duration of 20–200 ms. Events ± 30 ms were merged. Events were aligned to the nearest trough in the broadband data.

**Detector 5.** Data were band-pass filtered (80–120 Hz; Butterworth), Hilbert-transformed to extract the absolute instantaneous amplitude, smoothed using a 10 Hz low-pass filter (Butterworth), and z-normalized. Candidate ripple events were detected when the Hilbert envelope exceeded the threshold of 2 SD above the mean for a duration of 20–200 ms. Candidate events with an amplitude between 2–4 SD, frequency > 80 Hz, and no overlap with IEDs ± 1 relative to the ripple trough were considered for analysis. Events were aligned to the nearest trough in the broadband data.

**IED detection.** IED detection was conducted based on established protocols[22]. Preprocessing included band-pass filtering between 25 and 80 Hz, extraction of the signal envelope using the absolute Hilbert transform, and z-normalization. Events were detected when the envelope exceeded a threshold of 2 SD for a duration of 20–100 ms, and were discarded when the raw signal envelope did not exceed 2 SD. IEDs in close temporal proximity (1 s) were concatenated.

## Data analyses

*Spectral decomposition of simulated data* (Fig. 1c) was conducted on segmented data (30 s segments; 70% overlap) using a Hanning window (from 2–128 Hz in 1 Hz steps with a ¼ octave frequency smoothing) as implemented by Fieldtrip (ft_freqanalysis). The spectral exponent of the power spectral densities was determined between 20 and 45 Hz in log-log space with spectral parameterization[36] using linear fits (aperiodic mode = fixed) as implemented by Fieldtrip (ft_freqanalysis). This fitting range aligns with previous studies[33,35,85], has been shown to dissociate regional state-dependent dynamics between the MTL and PFC[40], and avoids potential artifacts caused by spectral knees or line noise. Goodness-of-fit was assessed using $R^2$ values. Continuous estimates of the spectral exponent were conducted following segmentation of the data (Fig. 2a: 30 s segments, 90% overlap; Fig. 3a, d, h: 1 s segments, 90% overlap). Average spectral exponents per sleep stage (30 s segments, no overlap) were calculated per channel. Condition-average spectral exponents (Fig. 3e, i) were calculated by segmenting the data per condition (Visual Search Task: 2 s segments; Motor Task: 3 s segments) without overlap.

*Ripple grand averages* (Figs. 1b, e, 3c, g) were derived following ripple detection (see above). Broadband data were segmented relative to the ripple trough (± 2.5 s) and amplitudes were z-normalized relative to baseline (−2.5 to −2 s relative to the ripple trough).

*Event density over time* (Figs. 2a; 3d, h, Middle & Bottom) delineates the number of events per second (1 s moving window in .1 s steps). *Event density per condition* (Fig. 3e, i Bottom) shows the average event density per condition (Inter-Trial-Interval [ITI] or Task conditions). Average event density per exponent bin (ranging from −8 to 0 in .1 steps) across the night for events detected on recorded iEEG data or

simulated data matching spectral exponent per segment (Fig. 2e Left) and sleep stage (Fig. 2d). We subsequently determined the percentage of detected ripples within the detection noise floor as a function of the spectral exponent of 1/f noise (Fig. 2f) by dividing the number of ripples detected on experimental data by those detected on surrogate simulated data.

*Electrode coverage* (Figs. 2b, 3b, f) shows electrode positions in MNI space on a standardized brain. Note that each electrode is represented by a sphere whose diameter does not represent the actual size of the electrode. For the Sleep dataset, electrode localization was determined in native space. Following localization, electrodes were subset for the medial temporal lobe (MTL; CA1, CA3/dentate gyrus, subiculum, entorhinal cortex, perirhinal cortex, parahippocampal gyrus, and temporal pole) and prefrontal cortex (PFC; medial and lateral prefrontal, as well as orbitofrontal cortices). Electrode locations (Figs. 2b, 3b, f) were mirrored for visualization purposes only to be shown on a single hemisphere. For the Visual Search dataset, electrode locations were determined using Fieldtrip (generate_electable_v3). Analyses only included electrodes localized in the visual cortex as determined by the VTPM atlas (Fig. 3b, $N_{electrodes}$ = 34)[79]. For the Motor Task dataset, electrode coordinates were transformed from Talairach to MNI coordinates using Fieldtrip (tal2mni). Electrode locations were determined using Fieldtrip (generate_electable_v3). Analyses only included electrodes localized in the motor cortex as determined by the Brainnetome atlas (Fig. 3f, $N_{electrodes}$ = 99)[80].

*Distributions* (Fig. 2d, $N_{bins}$ = 200) reflect the average number of segments (30 s) per spectral exponent per channel over a night of sleep. Data are shown on group level, and separated for iEEG electrodes localized to the MTL or PFC.

**Statistical analyses.** Condition and data type contrasts (Fig. 2e, f) were compared at the pseudo-population level using repeated-measures ANOVAs (RM-ANOVA) and were supported by Linear Mixed Effects models (LME) wherein subject and channel were included as independent variables to control for subject-specific effects. Effect sizes are reported as partial eta-squared ($\eta^2$) for the RM-ANOVAs and as beta coefficients ($\beta$) for the LMEs. Spearman correlational analyses were conducted for the spectral exponent and ripple density (Figs. 2e, 3e, i) using conditional differences (i.e., Wake – SWS and Task – ITI, respectively). Similarly, follow-up partial correlations for the spectral exponent and ripple density were conducted using conditional differences, wherein surrogate ripple density was considered as a confounding variable.

### Reporting summary
Further information on research design is available in the Nature Portfolio Reporting Summary linked to this article.

## Data availability
The electrophysiological task data are available at https://searchworks.stanford.edu/view/zk881ps0522. Source data are provided with this paper.

## Code availability
Freely available algorithms used for analyses are indicated where applicable. All custom code for 1/f noise simulations and ripple detection algorithms are available on GitHub (https://github.com/helfrichlab/vanSchalkwijk_1-f-ripples_NatCommun)[86] (https://doi.org/10.5281/zenodo.17978351).

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

## Acknowledgments

This work was supported by the German Research Foundation, Emmy Noether Program (DFG HE8329/2-1; R.F.H.) and Walter Benjamin Program (DFG SCHA 2369/1-1; F.J.V.S.); the Hertie Foundation (Network for Excellence in Clinical Neuroscience; R.F.H.) and the Jung Foundation for Research and Science (Ernst Jung Career Advancement Award in Medicine; R.F.H.). We acknowledge support from the Open Access Publishing Fund of the University of Tübingen.

## Author contributions

R.F.H. designed the study. F.J.V.S. and R.F.H. acquired and pre-processed the data. F.J.V.S. and R.F.H. curated the data. F.J.V.S. conducted formal analysis. F.J.V.S. conducted statistical analyses. F.J.V.S. and R.F.H. drafted the manuscript. R.F.H. supervised the project. All authors commented on and edited the manuscript draft.

## Funding

## Competing interests

The authors declare no conflict of interest.
