## [Transparent Peer Review file · Nature Communications]

Aperiodic $1/f$ noise drives ripple activity in humans

Corresponding Author: Dr Frank van Schalkwijk

Version 0:

Reviewer comments:

Reviewer #1

(Remarks to the Author)

This manuscript addresses the important question of whether sharp-wave ripples detected in human intracranial EEG data result from noise. The analyses seek to determine how aperiodic $1/f^x$ noise components and choice of ripple detection algorithm affects ripple detection. This includes characterization of various factors (brain state, brain region, level of engagement) and their influence on underlying $1/f^x$ characteristics which drive ripple detection. The authors use three human iEEG datasets to address these questions. Recordings were taken from MTL and LPFC of epileptic patients receiving presurgical monitoring for intractable epilepsy during sleep, occipital cortex during a visual search task, and motor cortex during a motor task. These physiological recordings were used to create $1/f^x$ aperiodic noise matched simulated datasets with physiologically plausible ripple morphology. Analyses of slow wave sleep (SWS) and $1/f^x$ -matched simulated data by five ripple detector algorithms show that each detector algorithm reports differing ripple densities as a function of the spectral exponent of the simulated dataset. Subsequent analyses focused on the effect of physiological differences (brain region identity and brain state) on ripple detection with the understanding that these physiological differences affect $1/f^x$ activity. Comparisons of iEEG recordings taken from sleep and awake states in the MTL and LPFC demonstrated the proportion of ripples in the noise floor differed by both brain region and state. The highest proportion of ripples within the noise floor was 77% from MTL recordings during wakefulness. To further the point that these $1/f^x$ aperiodic noise characteristics drive ripple detection, analyses were performed using iEEG recordings from occipital and motor cortex. These regions should not have SWRs during the tasks that altered $1/f^x$. Nonetheless, task engagement modulated the spectral exponent and subsequently the $1/f^x$ noise which was related to the reported ripple density throughout the task. In total, these findings illustrate that $1/f^x$ aperiodic noise components which may vary depending on brain region, brain state, and cognitive engagement with a task, affect ripple detection.

This study and its findings present compelling evidence for the case that $1/f^x$ aperiodic noise affects ripple detection. This is highly significant for the interpretation of reported SWRs in human iEEG datasets, which have likely reflected little if anything more than noise. The analyses are logical, rigorously executed, and straightforward to interpret. I have no major concerns and few minor comments.

1. Although the manuscript attempts to walk the fine line of criticizing previous characterizations of ripple oscillations in the human iEEG literature without calling out any specific study, such analyses could be useful. That is, it would help readers to appreciate the significance of these findings if specific examples of how these detectors have been used by previous studies were mentioned and, ideally, if prior data could be re-analyzed in light of these findings.
2. If nothing else, some explicit discussion of relevance to previous recommendations for SWR detection (e.g., Liu et al. 2022) should be included.
3. The choice to report the results of detector 1 needs to be justified. There is variability among ripple detection algorithms making the choice of detector 1 appear spurious. Would it be possible to better summarize results from all tested detectors?
4. It could be useful to clarify the definition of $1/f^x$ as $1/f^\chi$ where $\chi = -\alpha$. The current indication is slightly confusing as it uses conventions from two different bodies of literature. A sufficient alternative would be to represent $1/f^x$ in slope form.
5. The presence of interictal epileptiform discharges is acknowledged in the methods for event detection, but how IEDs are detected is not explained. Moreover, it is not immediately obvious that IEDs are accounted for during the detection of candidate ripple events which may present as an important detail given that in some conditions, ripple detection is performed during SWS where these epileptic phenomena are common.
6. The authors should report the units of ripple density. It is suggested in the methods section, but ambiguous in the plots and main text.
7. Figures 1E and 1E need a legend to describe which color corresponds to each detector. This is difficult to discern given the current format, including caption.

Reviewer #2

(Remarks to the Author)

Summary:

This paper investigated hippocampal sharp-wave ripples (SWR), which have been implicated in cognition in previous literature and more recently identified during wake states. Specifically, the authors looked at the performance of five different ripple detection algorithms on recorded data, as well as simulated data generated from the $1/f^x$ relationship, in order to understand the impacts of aperiodic 'noise' on ripple detection. Despite differences in density of detected ripples across the detection algorithms, the density of identified ripples always scaled with the spectral exponents. Next, authors demonstrate both regional- and state-specific differences in aperiodic noise, highlighting the necessity for future investigations to consider this noise floor during ripple detection to avoid false positives. These results are replicated in two regions where ripples have not previously been identified. Again we see the rate of ripple detection vary as a function of aperiodic noise. In sum, these results suggest that ripples identified during wakefulness and cognitive performance are likely false positives, reflecting aperiodic activity, rather than true ripple oscillations.

This is likely to be a very widely read, impactful paper. The results are quite striking and compelling, and this will certainly agitate some folks. Well done!

Major Comments:

From the methods, it is not clear whether or not simulated data was generated from spectral exponents alone, or if ripples were also simulated. We're fairly certain that no ripples were injected into the simulations, but rather the detected ripples in the simulation are just due to the stochastic nature of high frequency events. If this is correct, we believe that the authors should make an even bigger deal of this point! It's of course unusual to include in the methods something that you did not do, but in this case we feel that clarification on this is important because the detection algorithm's performance depends on whether ripples were present. If ripples were not simulated and the detection algorithms still identified these timepoints, this significantly impacts the interpretation of ripple detection.

In the Introduction, the authors state that "We specifically tested if more ripples are evident in cortical regions that exhibit a higher spectral exponent (e.g., hippocampus and the medial temporal lobe; MTL) as compared to neocortical association areas (e.g., lateral prefrontal cortex; LPFC)." This and the following about ripples in different states, as written, are not evidence that ripple detection is impacted by aperiodic activity. Rather, they suggest that ripples might just show regional and/or state differences. We believe that the authors frame these decisions more strongly later in the paper, where they explain that these choices included regions where we don't have strong prior evidence for ripples (such as M1). The authors should lean on this framing more in the Introduction, as well.

Minor Comments:

* The authors state that the data were referenced to a bipolar montage. This is somewhat unclear, as bipolar montages are typically applied in intracranial recordings, but are uncommon in scalp EEG. Clarification on the referencing process for the scalp EEG is needed.

* The authors estimated the spectral exponent from the 20-45 Hz range. But we recommend that they perform a sensitivity analysis, re-running a few key statistical comparisons but using different frequency ranges, to show that their results are relatively robust to the spectral fitting range.

Reviewer #3

(Remarks to the Author)

Version 1:

Reviewer comments:

Reviewer #1

(Remarks to the Author)

The authors adequately addressed all of the issues I raised. I have no other substantial concerns.

Reviewer #2

(Remarks to the Author)

The authors have done a fine job responding to all of our comments. We have no further concerns, and we are excited to see this paper get published.

Reviewer #3

(Remarks to the Author)

Version 2:

Reviewer comments:

Reviewer #4

(Remarks to the Author)

Reviewer comments are highlighted in **bold** and our responses in **blue**. The relevant paragraph(s) that were modified or added to the revised manuscript are indented. Particularly important changes are highlighted in **red**.

Reviewer #1 (Remarks to the Author):

This manuscript addresses the important question of whether sharp-wave ripples detected in human intracranial EEG data result from noise. The analyses seek to determine how aperiodic $1/f^x$ noise components and choice of ripple detection algorithm affects ripple detection. This includes characterization of various factors (brain state, brain region, level of engagement) and their influence on underlying $1/f^x$ characteristics which drive ripple detection. The authors use three human iEEG datasets to address these questions. Recordings were taken from MTL and LPFC of epileptic patients receiving presurgical monitoring for intractable epilepsy during sleep, occipital cortex during a visual search task, and motor cortex during a motor task. These physiological recordings were used to create $1/f^x$ aperiodic noise matched simulated datasets with physiologically plausible ripple morphology. Analyses of slow wave sleep (SWS) and $1/f^x$ -matched simulated data by five ripple detector algorithms show that each detector algorithm reports differing ripple densities as a function of the spectral exponent of the simulated dataset. Subsequent analyses focused on the effect of physiological differences (brain region identity and brain state) on ripple detection with the understanding that these physiological differences affect $1/f^x$ activity. Comparisons of iEEG recordings taken from sleep and awake states in the MTL and LPFC demonstrated the proportion of ripples in the noise floor differed by both brain region and state. The highest proportion of ripples within the noise floor was 77% from MTL recordings during wakefulness. To further the point that these $1/f^x$ aperiodic noise characteristics drive ripple detection, analyses were performed using iEEG recordings from occipital and motor cortex. These regions should not have SWRs during the tasks that altered $1/f^x$. Nonetheless, task engagement modulated the spectral exponent and subsequently the $1/f^x$ noise which was related to the reported ripple density throughout the task. In total, these findings illustrate that $1/f^x$ aperiodic noise components which may vary depending on brain region, brain state, and cognitive engagement with a task, affect ripple detection.

This study and its findings present compelling evidence for the case that $1/f^x$ aperiodic noise affects ripple detection. This is highly significant for the interpretation of reported SWRs in human iEEG datasets, which have likely reflected little if anything more than noise. The analyses are logical, rigorously executed, and straightforward to interpret. I have no major concerns and few minor comments.

We thank the reviewer for the positive evaluation and are grateful for the helpful comments. Below, we address all remaining queries.

R1.1. Although the manuscript attempts to walk the fine line of criticizing previous characterizations of ripple oscillations in the human iEEG literature without calling out any specific study, such analyses could be useful. That is, it would help readers to appreciate the significance of these findings if specific examples of how these detectors have been used by previous studies were mentioned and, ideally, if prior data could be re-analyzed in light of these findings.

We thank the reviewer for their remarks, and have expanded the Discussion section with the following new paragraph.

Discussion (p. 14): In this study, we evaluated five different ripple detectors. These detectors, or close variants thereof, have been used to study hippocampal¹⁻⁴ as well as neocortical⁵⁻⁹ ripples during NREM sleep and wakefulness. In addition, hippocampal ripples were reported during memory encoding and retrieval¹⁰⁻¹³, wherein differences in ripple rate between remembered and forgotten items during encoding or retrieval provide strong behavioral evidence. In light of the here reported association of 1/f noise and ripple detection, it might be conceivable that ripples as previously reported mainly index altered excitability dynamics and local cortical activation. This in accordance with a large body of literature that shows that increased high-frequency band activity (sometimes termed broadband or high-gamma) precisely indexes local cortical activation¹⁴⁻¹⁶. One conceivable approach for future studies is to closely examine oscillatory signatures (such as ripples) as well as aperiodic markers to infer to which extent these two markers overlap or reflect dissociable signatures. In the context of memory formation, some evidence suggests that these reflect dissociable signatures¹⁷.

R1.2. If nothing else, some explicit discussion of relevance to previous recommendations for SWR detection (e.g., Liu et al. 2022) should be included.

Our findings provide further empirical support to address several challenges identified by Liu and colleagues¹⁸. Specifically, our results demonstrate how background activity (the 1/f noise structure) is a critical factor that should be taken into account. We have expanded the Discussion section to clarify this point, and emphasize how simulation-based approaches might enable estimating the detection noise floor to obtain a false-positive rate. We additionally discuss how our findings connect to previous recommendations for ripple detection.

Discussion (p. 13): The gold standard for SWR detection in rodents are high-density laminar recordings that enable the visual identification of the distinct sharp-wave component arising in CA3 and the ripple component in CA1¹⁹. Hence, most detection algorithms that are employed in rodents take advantage of detecting multiple waveform characteristics^{20,21}. As virtually all intracranial EEG recordings in the human brain are obtained from epileptic patients, it is not feasible to detect sharp wave components as these also hallmark interictal discharges. Hence, it proves challenging to detect ripple events based on visual inspection, as even experts often cannot reach consensus²². To overcome these challenges, it is critical to move towards a consensus for the recording, detection, and reporting of ripples¹⁸, for which the authors provided a series of recommendations. An important step is the comparison to ripples detected during NREM sleep. The present study demonstrates a challenging aspect of this control analysis, as even simulated data may contain physiologically-plausible ripple events that resemble the typical

morphology in the time and frequency domain (**Fig. 1**). A second recommendation is accounting for EEG background activity. We demonstrate how background noise characteristics differ between states as well as regions (**Fig. 2**), which has not been accounted for in previously employed detection approaches. Crucially, changes in background activity may also give rise to spurious ripple detections in the neocortex (**Fig. 3**). Such false positive ripple detections result from fixed amplitude and duration thresholds applied to the signal envelope of the band-pass filtered EEG recordings – a methodology utilized by many detection algorithms (c.f. **Table 1**). These amplitude thresholds are often based on the standard deviation of the background activity across the entire recording, thus, not accounting for possible differences between brain states. We now demonstrate how noise floors can be estimated per state and region, and offer an additional approach to establish a false-positive detection rate. Note that the choice of frequency band, ripple duration, and various other detection criteria constitute additional sources of detection bias that likely account for the variability across studies. Further recommendations, such as simultaneous recording of micro- and macro-LFP in the hippocampus, localization to hippocampal subfields and layers, and investigating the effect of anti-seizure medication on ripple occurrence and sleep, provide avenues for future studies to demonstrate how ripple detection can be improved.

R1.3. The choice to report the results of detector 1 needs to be justified. There is variability among ripple detection algorithms making the choice of detector 1 appear spurious. Would it be possible to better summarize results from all tested detectors?

We agree with the reviewer that the choice for utilizing detector 1 for the main analyses requires justification. We chose to utilize detector 1 for all main analysis as it implements many of the characteristics observed across all detectors (i.e., filtering frequency, amplitude threshold, and event duration; c.f. **Table 1**), and its detection profile on 1/f noise closely matched two other detectors (**Fig. 1D**; detectors 2 and 5). The detector additionally demonstrated a relatively low noise floor for ripple detection in the medial temporal lobe during NREM sleep (**Fig. 2F**).

We now demonstrate how the noise profile of each methodology affects ripple detection on simulated data with varying exponents for every detector (**Fig. S1**), as was originally only shown for detector 1 (**Fig. 3A**). This replication has been added to the Supplementary Materials.

The Results section now provides additional context and illustrate qualitatively similar results as previously observed for detector 1.

Fig. 1 | Detecting ripple oscillations in 1/f noise. ... (D) Ripple density (Hz) as a function of the spectral exponent of the simulated data (mean \pm SD; 1 h duration; 100 iterations) for five different ripple detectors (different colors).

Fig. S1 | Detecting ripple oscillations in variable 1/f noise simulations. Demonstration of how the noise profile of each detector affects ripple detection on simulated data with variable 1/f characteristics. *Upper row:* Simulated signal with varying exponents (black) per 3 second segments. *Middle row:* Ripple-band activity (grey line; 80-120 Hz; z-normalized). *Bottom row:* Ripple density (Hz) per detector across identical simulated datasets and iterations (100 iterations; mean \pm SEM).

Results (p. 5): Since all detection algorithms exhibit their own idiosyncrasies, we conducted all subsequent analyses using the first detector. This detector was chosen for the main analyses as it utilizes several characteristics shared across detectors (c.f., Table 1) and yields a representative detection profile on 1/f noise (c.f., Fig. 1D; SI Appendix, Fig. S1). Note that qualitatively and quantitatively similar results were observed for the other detectors (SI Appendix).

Results (p. 6-7): Collectively, this set of findings demonstrates that region- and state-specific differences in 1/f noise characteristics severely impact ripple detection, especially if the detector noise susceptibility (Fig. 1D) peaks within the experimental condition as exemplified for MTL ripples. While the spectral exponent of the MTL during wakefulness matched the noise profile, the spectral exponent was lower during SWS and outside of the detector noise range; thus, explaining why more false positive ripples were detected during wakefulness as compared to SWS. Note that different detection algorithms exhibit varying detection patterns on 1/f noise, but give rise to qualitatively comparable results (SI Appendix Fig. S1; Fig. S3).

Results (p. 9): To demonstrate the feasibility of this approach, we first simulated 1/f noise with varying exponents over several seconds that might correspond to different task epochs (Fig. 3A; SI Appendix Fig. S1). As expected, ripple detection was systematically biased to the state with the highest spectral exponent. This bias was driven by the predefined amplitude threshold criterion in the ripple detection algorithm. This fixed threshold is derived from the mean and variance of the band-pass-filtered signal envelope across the entire recording; a methodological approach that is common to most detection algorithms (cf. Table 1).

We additionally acknowledge that the main analyses had to be replicated for all detectors, as this further emphasizes how ripple detection is affected by the noise distribution of the data. Here, the noise floor of ripple detection is defined as the percentage of detected ripples on simulated data relative to those detected on simulated surrogate datasets. We have replicated our main finding for the Sleep dataset, wherein we show ripple density and noise floor estimates across wakefulness and slow-wave sleep (SWS) for detector 1 (Fig. 2E-F), and demonstrate that this pattern can be observed for all detectors as outlined

in the revised supplemental material (see below; **Fig. S2**). Note that despite the different detection characteristics (c.f., **Fig. 1D**), similar detection patterns were observed for the other detectors (**Fig. S2C**).

To make these results accessible, we now provide a summary in the main manuscript (see below: **Fig. 2G**), color-coding the median values to demonstrate similar state and regional changes in the noise floor per detector. We also provide all details in the supplements (**Fig. S3**).

Results (p. 6): Similar effects can be observed for all other detectors (**Fig. 2G; SI Appendix Fig. S3**). In sum, these results indicate state- and region- dependent modulation of ripple density that follows a $1/f^k$ relationship.

Fig. 2 | Different $1/f$ noise characteristics explain region- and state-specific ripple variability. ... (G) Noise-floor estimates across states and regions for all five ripple detectors. Note that the majority of detection algorithms demonstrate the same state and regional differences. Colors depict median values per detector, state, and region (wherein the first row corresponds to the values displayed in panel *F*).

Fig. S3 | 1/f noise characteristics variably explain region- and state-specific ripple density as a function of detection algorithm. (A) Region- and state-specific relationship between noise and ripples. *Left:* Distribution of ripple events (y-axis) as a function of the spectral exponent (x-axis) in the medial temporal lobe (MTL; red) per detection algorithm, highlighting a clear relationship similar to the distribution reported in Figure 1D. Ripple detection in noise-matched epochs (superimposed grey distribution) highlights MTL ripples within the noise range and therefore, likely do not constitute genuine ripple events. *Right:* Distribution of ripple events relative to the spectral exponent in the lateral prefrontal cortex (LPFC; same conventions as in the left panels). **(B)** Mean event density per state. *Left:* We observed more ripples in the MTL during wakefulness than during SWS, as a direct result of higher spectral exponents during wakefulness (cf. top panel Fig. 2C). Given the overall lower exponents during SWS, it is more likely to detect true ripples during SWS than during wakefulness. Boxplots represent median, 1st and 3rd quartiles, and extreme values. *Right:* Mean event density per state in the LPFC. Same conventions as in the left panels. **(C)** Percentage of ripples within the noise floor. Note that especially ripple detection in the MTL during wakefulness exhibit high percentages of false positives. Same conventions as in panel B.

Results (p. 6): To illustrate and summarize these results, we calculated the percent overlap of observed ripples relative to the surrogate distribution (**Fig. 2F**). In the MTL, we observed that the median proportion of wake ripples falling within the noise floor was 77%, while only 23% during SWS could be attributed to the $1/f$ characteristics. In the LPFC, 4% of ripple detections during wakefulness and 28% during SWS possibly reflected false positives. Statistical quantification confirmed that false positive ripple detections significantly decreased from wakefulness to SWS in the MTL ($F_{1,81} = 21.16$, $p < .0001$, $\eta^2 = .21$, RM-ANOVA; $p < .0001$, $[CI_{95}] = [-37.33, -15.0]$, $t_{162} = -4.63$, $\beta = -26.17$, LME), but showed the reversed pattern in the LPFC ($F_{1,338} = 122.98$, $p < .0001$, $\eta^2 = .27$, RM-ANOVA; $p < .0001$, $[CI_{95}] = [17.7, 25.3]$, $t_{676} = 11.11$, $\beta = 21.54$, LME). We consequently observed a significant interaction effect between the two regions ($F_{1,419} = 97.28$, $p < .0001$, $\eta^2 = .19$, RM-ANOVA; $p < .0001$, $[CI_{95}] = [38.24, 57.18]$, $t_{838} = 9.89$, $\beta = 47.71$, LME). **Similar effects can be observed for all other detectors (Fig. 2G; SI Appendix Fig. S3)**. In sum, these results indicate state- and region- dependent modulation of ripple density that follows a $1/f^\alpha$ relationship.

R1.4. It could be useful to clarify the definition of $1/f^\alpha x$ as $1/f^\alpha x$ where $\alpha = -\alpha$. The current indication is slightly confusing as it uses conventions from two different bodies of literature. A sufficient alternative would be to represent $1/f^\alpha x$ in slope form.

We acknowledge that this description caused confusion, and have added an explicit statement wherein we define $1/f$ noise.

Introduction (p. 2): In a largely unrelated line of inquiry, it had recently been shown that electrophysiological ‘background’ activity, also termed aperiodic activity for the lack of a unique periodicity, indexes different brain states and cognitive demands. Aperiodic activity is also called $1/f$ noise as it follows a f^α scaling law. **We defined $1/f$ noise as $1/f^\alpha$, which yielded a positive exponent. We adopted the common convention to report the exponent in slope form as the negative exponent**

23,24.

R1.5. The presence of interictal epileptiform discharges is acknowledged in the methods for event detection, but how IEDs are detected is not explained. Moreover, it is not immediately obvious that IEDs are accounted for during the detection of candidate ripple events which may present as an important detail given that in some conditions, ripple detection is performed during SWS where these epileptic phenomena are common.

The detection of interictal epileptiform discharges (IEDs) and removal of ripple candidates in close temporal proximity are important topics that were not fully addressed in the manuscript. We have now added the methodology on IED detection to the Methods section.

Methods (p. 23): *IED detection.* IED detection was conducted based on established protocols²⁵. Preprocessing included band-pass filtering between 25 and 80 Hz, extraction of the signal envelope using the absolute Hilbert transform, and z-normalization. Events were detected when the envelope exceeded a threshold of 2 SD for a duration of 20-100 ms, and were discarded when the raw signal envelope did not exceed 2 SD. IEDs in close temporal proximity (1 s) were concatenated.

The reviewer correctly points out that epileptic phenomena are common in our experimental datasets. It should therefore be clear that IED detection was conducted on all experimental datasets, and that candidate ripple events in close temporal proximity to IEDs (± 2.5 s) were discarded. We acknowledge that this detail has to be emphasized in both Methods and Results sections, as it was only reported in the detector methodology (Methods). We have revised the Results section to make this correction step more apparent.

Results (p. 5): A separate detection of interictal epileptiform discharges (IEDs) was conducted for the experimental datasets (Methods), thus, discarding any candidate ripple events in close temporal proximity of an IED (± 2.5 s); thereby minimizing erroneously considering IEDs as ripples.

Results (p. 9): We hypothesized that ripple detection should reflect the 1/f modulation through increased ripple detections during the task epoch. Moreover, we reasoned that if this ripple increase solely stems from a change in 1/f noise, then ripple detection on simulated, noise-matched surrogates should yield a comparable number of ripples. Ripple detection was conducted identically to our analysis on the *Sleep* dataset, and ripples in close temporal proximity to IEDs (± 2.5 s) were discarded.

R1.6. The authors should report the units of ripple density. It is suggested in the methods section, but ambiguous in the plots and main text.

Ripple density is defined as the number of events per second, and can therefore be reported in Hz. The original figures utilized the symbol ρ , which we have now replaced with Hz as a more accurate report. We have revised the text and figures of the main manuscript as well as the Supplementary Materials.

Results (p. 4): We observed that the density of detected ripples (defined as the number of events per second; Hz) differed between algorithms, but always scaled with the spectral exponent (Fig. 1D; all $p < 0.0001$, RM-ANOVA; effect sizes: detector (1): $\eta^2 = 0.99$; (2): $\eta^2 = 0.96$; (3): $\eta^2 = 0.95$; (4): $\eta^2 = 0.08$; (5): $\eta^2 = 0.99$).

R1.7. Figures 1E and 1E need a legend to describe which color corresponds to each detector. This is difficult to discern given the current format, including caption.

The reviewer correctly points out that it was not possible to identify the ripple detectors based on color in Figures 1D and 1E. We added a legend to Fig. 1D.

Reviewer #2 (Remarks to the Author):

Summary:

This paper investigated hippocampal sharp-wave ripples (SWR), which have been implicated in cognition in previous literature and more recently identified during wake states. Specifically, the authors looked at the performance of five different ripple detection algorithms on recorded data, as well as simulated data generated from the $1/f^x$ relationship, in order to understand the impacts of aperiodic 'noise' on ripple detection. Despite differences in density of detected ripples across the detection algorithms, the density of identified ripples always scaled with the spectral exponents. Next, authors demonstrate both regional- and state-specific differences in aperiodic noise, highlighting the necessity for future investigations to consider this noise floor during ripple detection to avoid false positives. These results are replicated in two regions where ripples have not previously been identified. Again we see the rate of ripple detection vary as a function of aperiodic noise. In sum, these results suggest that ripples identified during wakefulness and cognitive performance are likely false positives, reflecting aperiodic activity, rather than true ripple oscillations. This is likely to be a very widely read, impactful paper. The results are quite striking and compelling, and this will certainly agitate some folks. Well done!

We thank the reviewer for the positive evaluation of our manuscript.

Major Comments:

R2.1. From the methods, it is not clear whether or not simulated data was generated from spectral exponents alone, or if ripples were also simulated. We're fairly certain that no ripples were injected into the simulations, but rather the detected ripples in the simulation are just due to the stochastic nature of high frequency events. If this is correct, we believe that the authors should make an even bigger deal of this point! It's of course unusual to include in the methods something that you did not do, but in this case we feel that clarification on this is important because the detection algorithm's performance depends on whether ripples were present. If ripples were not simulated and the detection algorithms still identified these timepoints, this significantly impacts the interpretation of ripple detection.

We thank the reviewer for this excellent remark. Indeed, the simulated data were generated solely based on the spectral exponents derived from the experimental datasets. The reviewer correctly identifies that no ripples were added to the simulations. We are appreciative of the encouragement, and have revised the Methods, Results, and Discussion sections to emphasize this point.

Methods (p. 22): Colored noise were simulated using the "*colored_noise*" function²⁶. Here, white noise is filtered to create colored noise characterized with a defined spectral exponent of its power spectrum density. Data were iteratively simulated ($N = 100$) for a range of spectral exponents (-4 to 0 in .1 steps) at 1000 Hz, followed by minimum standard EEG preprocessing steps where the data were demeaned, detrended, band-pass filtered (.5 – 150 Hz), and z-normalized. Data were simulated with a constant spectral exponent (**Fig. 1A**) or approximated the spectral

exponent of the recorded iEEG data per segment over time (**Fig. 2A**: 30 s segments without overlap; **Fig. 3A, D, H**; 1 s segments without overlap). Simulated data were created per segment per channel and subsequently concatenated. **It is critical to highlight that no simulated ripples were added to the surrogate datasets.**

Results (p. 3): To quantify the relationship between $1/f$ noise and ripple detections, we simulated multiple one-hour long surrogate EEG traces that followed a f^{χ} relationship with varying exponents χ , ranging from -4.0 to 0 in 0.1 steps. **Note that the surrogate datasets ($1/f$ noise time series) were simulated solely based on the pre-defined spectral exponent. Hence, we did not simulate ripple oscillations on top of the $1/f$ noise, indicating that all detections were solely obtained from the noise time series devoid of any true oscillatory activity.**

Discussion (p. 11-12): In simulations and intracranial human recordings across three experiments, we demonstrate that putative ripples largely result from different $1/f$ noise characteristics that systematically vary as a function of cortical region, state, and task demands. **It is critical to highlight that all detection algorithms identified ripples for simulated datasets that solely consisted of colored noise with pre-defined spectral exponents.** Our results provide a framework based on surrogate $1/f$ noise simulations to infer a state-specific false positive rate. Moreover, these findings reveal the noise characteristics of various detection algorithms and highlight the need to account for context-dependent $1/f$ activity during ripple detection.

R2.2. In the Introduction, the authors state that “We specifically tested if more ripples are evident in cortical regions that exhibit a higher spectral exponent (e.g., hippocampus and the medial temporal lobe; MTL) as compared to neocortical association areas (e.g., lateral prefrontal cortex; LPFC).” This and the following about ripples in different states, as written, are not evidence that ripple detection is impacted by aperiodic activity. Rather, they suggest that ripples might just show regional and/or state differences. We believe that the authors frame these decisions more strongly later in the paper, where they explain that these choices included regions where we don’t have strong prior evidence for ripples (such as M1). The authors should lean on this framing more in the Introduction, as well.

It is correct to state that our investigation for ripple occurrence in different states and cortical regions in the experimental data does not directly provide evidence that ripple detection is impacted by aperiodic activity. The Introduction section now highlights our initial analyses on simulated datasets, as well as the final analyses conducted during wakefulness, wherein we evaluated state-specific changes in ripple detection in light of the $1/f$ noise characteristics of background activity. These analyses were conducted on electrodes in the early visual and motor cortex – regions where one would not expect ripples.

Introduction (p. 2-3): To address this hypothesis, we combined three experiments conducted during sleep and wakefulness with noise simulations to test if, and to which extent, noise characteristics impact ripple detection. **Using simulated 1/f noise with varying spectral exponents, we tested if ripple detection is modulated as a function of the underlying 1/f noise.** In a next step using experimental data, we specifically tested if more ripples are evident in cortical regions that exhibit a higher spectral exponent (e.g., hippocampus and the medial temporal lobe; MTL) as compared to neocortical association areas (e.g., lateral prefrontal cortex; LPFC). In addition, we determined the relationship of 1/f noise and ripples during different cortical states (i.e., sleep, quiet wakefulness, and cognitive engagement). Given prominent, state-dependent 1/f differences wherein the exponents are high during cognitive engagement and lower during sleep, we predicted a higher ripple rate in the wake state as a result of increased 1/f noise. **To test if state-specific changes in ripple detection reflect an altered 1/f noise characteristic, we conducted analyses on electrodes in the early visual and motor cortex – regions for which no strong prior evidence of ripple occurrence exists. We reasoned that ripple detections should be driven by the previously documented change in 1/f noise** ^{27,28}.

Minor Comments:

R2.3. The authors state that the data were referenced to a bipolar montage. This is somewhat unclear, as bipolar montages are typically applied in intracranial recordings, but are uncommon in scalp EEG. Clarification on the referencing process for the scalp EEG is needed.

We acknowledge that the preprocessing of the scalp EEG was not clearly defined. Rereferencing to a bipolar montage was only applied in the preprocessing of the intracranial EEG data, whereas the simultaneously acquired scalp EEG was rereferenced to the common average over all available scalp electrodes. The differentiation in data preprocessing has now been made more explicit, and additional information on preprocessing of the scalp EEG data has been added to the Methods section.

Methods (p. 21): Intracranial EEG data from the *Sleep* dataset were demeaned, detrended, rereferenced to a bipolar montage, and downsampled to 500 Hz. Simultaneously recorded scalp EEG data utilized for sleep scoring were demeaned, detrended, referenced to the common average over all scalp electrodes, low-pass filtered (50 Hz), and downsampled to 500 Hz. Data from the *Visual Search Task* and *Motor Task* were rereferenced to the common average.

R2.4. The authors estimated the spectral exponent from the 20-45 Hz range. But we recommend that they perform a sensitivity analysis, re-running a few key statistical comparisons but using different frequency ranges, to show that their results are relatively robust to the spectral fitting range.

Our results demonstrate that the spectral exponent of the power spectral density, determined between 20 and 45 Hz in log-log space with spectral parameterization ²⁹, identifies region-, state-, and demand-dependent modulation of cortical background activity (**Fig. 2; Fig.3**). This specific frequency range was chosen as it is generally devoid of narrow-banded oscillations, follows a clear 1/f decay, avoids line noise in both Europe and the US, and has been shown to reliably differentiate cortical states and regions ³⁰. Moreover, this frequency range substantially overlaps with the canonical low gamma range that may track cortical excitability ³¹.

As no consensus has been established with regard to the optimal frequency range for spectral estimation, we repeated the analyses for the sleep data (**Fig. 2**) using different frequency ranges for spectral parameterization (1-40 Hz; 20-45 Hz; 30-100 Hz; and 2-128 Hz). All models utilized a linear fit (aperiodic mode = fixed).

Model fits showed high goodness-of-fit estimates ($R^2 \geq .97$), wherein the best fit was observed when the full spectrum was considered (1-40 Hz: $R^2 = .9830 \pm .0004$, mean \pm SEM; 20-45 Hz: $R^2 = .9817 \pm .0008$; 30-100 Hz: $R^2 = .9846 \pm .0013$; 2-128 Hz: $R^2 = .9908 \pm .0002$). Spectral exponents from the original fit (20-45 Hz) were significantly correlated with the 1-40 Hz range (all $p \leq .0001$; $\rho = .09 \pm .05$ mean \pm SEM), 30-100 Hz range (all $p \leq .0001$; $\rho = .36 \pm .04$), and the 2-128 Hz range (all $p \leq .0004$; $\rho = .25 \pm .05$).

Although all model fits differentiate between sleep states (**Fig. S2A**), the 2-128 Hz range did not capture the regional differences observed for the narrower frequency range and showed an overall narrower distribution of spectral exponents across the night (**Fig. S2B**) as also demonstrated recently³⁰. Note that all models can be employed to estimate a noise floor for ripple detection (**Fig. S2B**), albeit that the narrower distribution for the 2-128 Hz range results in the fact that all detected ripples fall within the noise floor boundaries. Combined, these simulations reveal a general pattern where ripple detection on experimental data has a wider distribution of exponent values as compared to simulated data.

In sum, our original approach (linear FOOOF model fit between 20-45 Hz) provides a very high goodness-of-fit ($R^2 = .9817 \pm .0008$; mean \pm SEM), omits the spectral knees in low and high frequency ranges, avoids line noise or notch filtering artifacts, delineates regional differences, and can be successfully utilized to identify the noise floor for ripple detection. These findings are referred to in the main manuscript.

Fig. S2 | State- and regional differences for spectral exponent values depends on the model fit. (A) Spectral exponents (y-axis) as a function of state (x-axis) and region (medial temporal lobe [MTL], red; lateral prefrontal cortex [LPFC], blue). Data are displayed as probability density functions. Model fits within the 20-45 Hz frequency range successfully differentiate between cognitive states and additionally capture regional differences. In contrast, a model fit across the entire spectrum (2-128 Hz) only differentiates between states. **(B)** Distribution of ripple density (y-axis) as a function of the spectral exponent (x-axis) in the MTL (red) and LPFC (blue). Simulation-based inference of ripple density (superimposed grey distribution) can be established regardless of the model's frequency range. Note how the distribution of spectral exponents is narrower for the model fit across the entire spectrum (*Bottom*) as compared to the alternative fits (*Top & Middle*), resulting in all events falling within the noise floor boundaries.

Results (p. 5-6): Qualitatively and quantitatively similar results were evident in the LPFC (**Fig. 2E Bottom Left**), where 28% of ripples were within the noise floor (exponent range: -2.7 to -.4), as well as for different spectral parametrization approaches (*SI Appendix Fig. S2*).

To determine the robustness of our findings, we repeated spectral parameterization of our wake datasets (**Fig. 3D, H, Top**) using multiple spectral fitting ranges (1-40 Hz; 20-45 Hz; 30-100 Hz; and 2-128 Hz). Again, condition-specific changes of the spectral exponent values were assessed.

For the *Visual Search Task*, we demonstrate that the spectral exponent increases during the task epoch as compared to inter-trial intervals (ITIs; **Fig. S5A**). Within the time domain, the spectral exponent increased during task epochs as compared to the inter-trial interval (ITI) (**Fig. S5A**). Statistical quantification was conducted at the pseudo-population level using RM-ANOVAs, and now includes frequency range as an additional factor, demonstrating significant main effects of condition (**Fig. S5B**; $F_{2,66} = 7.82$, $p = .0085$, $\eta^2 = .19$, RM-ANOVA) and frequency range ($F_{3,99} = 21.73$, $p < .0001$, $\eta^2 = .40$, RM-ANOVA), as well as an interaction effect between condition and frequency range ($F_{6,198} = 4.92$, $p = .0114$, $\eta^2 = .13$, RM-ANOVA).

A qualitatively and quantitatively highly comparable set of findings was observed for the *Motor Task*. In the time domain, the spectral exponent increases during the task epoch as compared to inter-trial intervals (ITIs) (**Fig. S5C**). Statistical quantifications demonstrate significant main effects of condition (**Fig. S5B**; $F_{2,196} = 105.51$, $p < .0001$, $\eta^2 = .52$, RM-ANOVA) and frequency range ($F_{3,294} = 339.60$, $p < .0001$, $\eta^2 = .78$, RM-ANOVA), as well as an interaction effect between condition and frequency range ($F_{6,588} = 58.67$, $p < .0001$, $\eta^2 = .37$, RM-ANOVA).

In sum, these effects show that the spectral exponent is modulated by task condition irrespective of frequency range used for spectral decomposition. Yet, the magnitude of the effect differs between frequency ranges, and depends on the model parameters used for spectral parameterization. Specifically, the frequency range considered, as well as the inclusion of a spectral knee, may account for the current variability^{24,32}. These findings have been added to the Supplementary Information (**Fig. S5**) and are referred to in the Results section.

Fig. S5 | Task-locked modulation of 1/f spectral exponent can be observed for multiple frequency range. (A) Time-resolved spectral exponent values per frequency range (colored traces) during execution of a visual search task. Note the modulation during task execution as compared to the pre and post inter-trial intervals (ITI) per frequency range considered for spectral parameterization. **(B)** Statistical quantification. The spectral exponent (median ± SEM) significantly increases during the task epoch as compared to the ITI. Same conventions as in panel A. Offset of the x-axis is for visualization purposes only. **(C-D)** Noise modulation during movement execution (same conventions as in panels A-B).

Results (p. 10): Supplementary analyses demonstrate that modulation of the spectral exponent by task is irrespective of the frequency range used for spectral parameterization (*SI Appendix Fig. S5A-B*).

Results (p. 10): Additional analyses demonstrate that modulation of the spectral exponent by task is irrespective of the frequency range used for spectral parameterization (*SI Appendix Fig. S5C-D*).

Reviewer #3 (Remarks to the Author):

We thank the reviewer for their co-review of our work and the helpful comments.

References

1. Axmacher, N., Elger, C. E. & Fell, J. Ripples in the medial temporal lobe are relevant for human memory consolidation. *Brain* **131**, 1806–1817 (2008).
2. Clemens, Z. *et al.* Temporal coupling of parahippocampal ripples, sleep spindles and slow oscillations in humans. *Brain* **130**, 2868–2878 (2007).
3. Ngo, H.-V., Fell, J. & Staresina, B. Sleep spindles mediate hippocampal-neocortical coupling during long-duration ripples. *eLife* **9**, e57011 (2020).
4. Staresina, B. P. *et al.* Hierarchical nesting of slow oscillations, spindles and ripples in the human hippocampus during sleep. *Nat Neurosci* **18**, 1679–1686 (2015).
5. Dickey, C. W. *et al.* Cortical Ripples during NREM Sleep and Waking in Humans. *J. Neurosci.* **42**, 7931–7946 (2022).
6. Dickey, C. W. *et al.* Widespread ripples synchronize human cortical activity during sleep, waking, and memory recall. *Proceedings of the National Academy of Sciences* **119**, e2107797119 (2022).
7. van Schalkwijk, F. J. *et al.* An evolutionary conserved division-of-labor between archicortical and neocortical ripples organizes information transfer during sleep. *Progress in Neurobiology* **227**, 102485 (2023).
8. Vaz, A. P., Inati, S. K., Brunel, N. & Zaghoul, K. A. Coupled ripple oscillations between the medial temporal lobe and neocortex retrieve human memory. *Science* **363**, 975–978 (2019).
9. Vaz, A. P., Wittig, J. H., Inati, S. K. & Zaghoul, K. A. Replay of cortical spiking sequences during human memory retrieval. *Science* **367**, 1131–1134 (2020).
10. Henin, S. *et al.* Spatiotemporal dynamics between interictal epileptiform discharges and ripples during associative memory processing. *Brain* **144**, 1590–1602 (2021).
11. Kunz, L. *et al.* Ripple-locked coactivity of stimulus-specific neurons and human associative memory. *Nat Neurosci* **27**, 587–599 (2024).
12. Norman, Y. *et al.* Hippocampal sharp-wave ripples linked to visual episodic recollection in humans. *Science* **365**, eaax1030 (2019).
13. Zhang, H. *et al.* Awake ripples enhance emotional memory encoding in the human brain. *Nat Commun* **15**, 215 (2024).
14. Crone, N. E., Miglioretti, D. L., Gordon, B. & Lesser, R. P. Functional mapping of human sensorimotor cortex with electrocorticographic spectral analysis. II. Event-related synchronization in the gamma band. *Brain* **121**, 2301–2315 (1998).
15. Edwards, E., Soltani, M., Deouell, L. Y., Berger, M. S. & Knight, R. T. High Gamma Activity in Response to Deviant Auditory Stimuli Recorded Directly From Human Cortex. *Journal of Neurophysiology* **94**, 4269–4280 (2005).
16. Miller, K. J., Zanos, S., Fetz, E. E., Den Nijs, M. & Ojemann, J. G. Decoupling the Cortical Power Spectrum Reveals Real-Time Representation of Individual Finger Movements in Humans. *J. Neurosci.* **29**, 3132–3137 (2009).
17. Fellner, M.-C. *et al.* Spectral fingerprints or spectral tilt? Evidence for distinct oscillatory signatures of memory formation. *PLoS Biology* **17**, e3000403 (2019).
18. Liu, A. A. *et al.* A consensus statement on detection of hippocampal sharp wave ripples and differentiation from other fast oscillations. *Nat Commun* **13**, 6000 (2022).
19. Buzsáki, G. Hippocampal sharp wave-ripple: A cognitive biomarker for episodic memory and planning. *Hippocampus* **25**, 1073–1188 (2015).
20. Fernández-Ruiz, A. *et al.* Long-duration hippocampal sharp wave ripples improve memory. *Science* **364**, 1082–1086 (2019).
21. Oliva, A., Fernández-Ruiz, A., Buzsáki, G. & Berényi, A. Role of Hippocampal CA2 Region in Triggering Sharp-Wave Ripples. *Neuron* **91**, 1342–1355 (2016).
22. Maslarova, A. *et al.* Spatiotemporal Patterns Differentiate Hippocampal Sharp-Wave Ripples from Interictal Epileptiform Discharges in Mice and Humans. Preprint at <https://doi.org/10.1101/2025.02.06.636758> (2025).
23. Colombo, M. A. *et al.* The spectral exponent of the resting EEG indexes the presence of consciousness during unresponsiveness induced by propofol, xenon, and ketamine. *NeuroImage* **189**, 631–644 (2019).
24. Donoghue, T. A Systematic Review of Aperiodic Neural Activity in Clinical Investigations. *European Journal of Neuroscience* **62**, e70255 (2025).
25. Gelinás, J. N., Khodagholy, D., Thesen, T., Devinsky, O. & Buzsáki, G. Interictal epileptiform discharges induce hippocampal–cortical coupling in temporal lobe epilepsy. *Nat Med* **22**, 641–648 (2016).
26. Das, A. Camouflage Detection & Signal Discrimination: Theory, Methods & Experiments. (2022). doi:10.13140/RG.2.2.10585.80487.
27. Miller, K. J. *et al.* Dynamic Modulation of Local Population Activity by Rhythm Phase in Human Occipital Cortex During a Visual Search Task. *Front. Hum. Neurosci.* **4**, 197 (2010).

28. Miller, K. J. *et al.* Spectral Changes in Cortical Surface Potentials during Motor Movement. *J. Neurosci.* **27**, 2424–2432 (2007).
29. Donoghue, T. *et al.* Parameterizing neural power spectra into periodic and aperiodic components. *Nat Neurosci* **23**, 1655–1665 (2020).
30. Lendner, J. D., Lin, J. J., Larsson, P. G. & Helfrich, R. F. Multiple intrinsic timescales govern distinct brain states in human sleep. *J. Neurosci.* **44**, e0171242024 (2024).
31. Gao, R., Peterson, E. J. & Voytek, B. Inferring synaptic excitation/inhibition balance from field potentials. *NeuroImage* **158**, 70–78 (2017).
32. Ameen, M. S., Jacobs, J., Schabus, M., Hoedlmoser, K. & Donoghue, T. *The Temporal Dynamics of Aperiodic Neural Activity Track Changes in Sleep Architecture.* <http://biorxiv.org/lookup/doi/10.1101/2024.01.25.577204> (2024) doi:10.1101/2024.01.25.577204.